# MoVie: Visual Model-Based Policy Adaptation for View Generalization

**Sizhe Yang**[12*], **Yanjie Ze**[13*], **Huazhe Xu**[415]

[1]Shanghai Qi Zhi Institute, [2]University of Electronic Science and Technology of China,
[3]Shanghai Jiao Tong University, [4]Tsinghua University, [5]Shanghai AI Lab
[*] Equal contribution
yangsizhe.github.io/MoVie

## Abstract

Visual Reinforcement Learning (RL) agents trained on limited views face significant challenges in generalizing their learned abilities to unseen views. This inherent difficulty is known as the problem of *view generalization*. In this work, we systematically categorize this fundamental problem into four distinct and highly challenging scenarios that closely resemble real-world situations. Subsequently, we propose a straightforward yet effective approach to enable successful adaptation of visual **Mo**del-based policies for **Vie**w generalization (**MoVie**) during test time, without any need for explicit reward signals and any modification during training time. Our method demonstrates substantial advancements across all four scenarios encompassing a total of **18** tasks sourced from DMControl, xArm, and Adroit, with a relative improvement of **33**%, **86**%, and **152**% respectively. The superior results highlight the immense potential of our approach for real-world robotics applications. Code and videos are available at yangsizhe.github.io/MoVie.

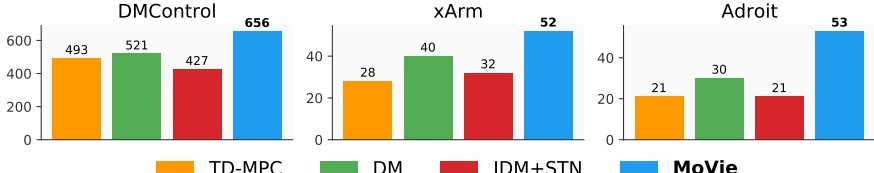

Figure 1: **View generalization results across 3 domains.** Our method MoVie largely improves the test-time view generalization ability, especially in robotic manipulation domains.

## 1  Introduction

Visual Reinforcement Learning (RL) has achieved great success in various applications such as video games [17, 18], robotic manipulation [22], and robotic locomotion [34]. However, one significant challenge for real-world deployment of visual RL agents remains: a policy trained with very limited views (commonly one single fixed view) might not generalize to unseen views. This challenge is especially pronounced in robotics, where a few fixed views may not adequately capture the variability of the environment. For instance, the RoboNet dataset [3] provides diverse views across a range of manipulation tasks, but training on such large-scale data only yields a moderate success rate ($10\% \sim 20\%$) for unseen views [3].

Recent efforts have focused on improving the visual generalization of RL agents [1,7,9,36]. However, these efforts have mainly concentrated on generalizing to different appearances and backgrounds. In contrast, view generalization presents a unique challenge as the deployment view is unknown and may move freely in the 3D space. This might weaken the weapons such as data augmentations [12, 14,15,33] that are widely used in appearance generalization methods. Meanwhile, scaling up domain

37th Conference on Neural Information Processing Systems (NeurIPS 2023), New Orleans.

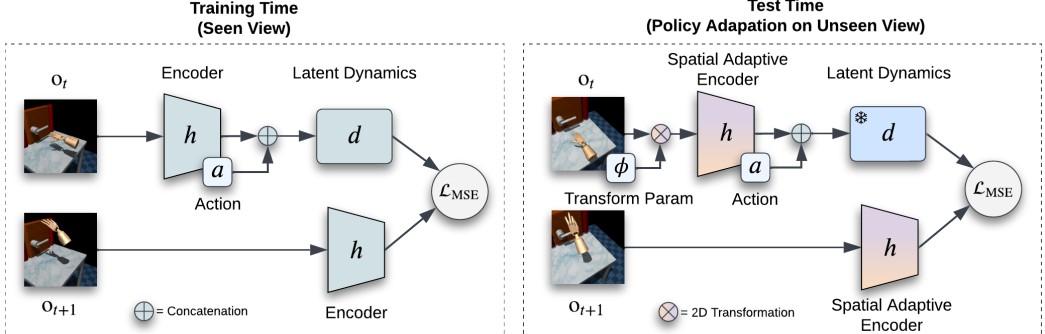

Figure 2: **Overview of MoVie**. During training (left), the agent is trained with the latent dynamics loss. At test time (right), we freeze the dynamics model and modify the encoder as a spatial adaptive encoder to adapt the agents to novel views.

randomization [19, 21, 31] to all possible views is usually unrealistic because of the large cost and the offline nature of existing robot data. With these perspectives combined, it is difficult to apply those common approaches to address view generalization.

In this work, we commence by explicitly formulating the test-time view generalization problem into four challenging settings: *a)* **novel view**, where the camera changes to a fixed novel position and orientation; *b)* **moving view**, where the camera moves continuously around the scene, *c)* **shaking view**, where the camera experiences constant shaking, and *d)* **novel FOV**, where the field of view (FOV) of the camera is altered initially. These settings cover a wide spectrum of scenarios when visual RL agents are deployed to the real world. By introducing these formulations, we aim to advance research in addressing view generalization challenges and facilitate the utilization of robot datasets [3] and deployment on physical robotic platforms.

To address the view generalization problem, we argue that *adaptation* to novel views during test time is crucial, rather than aiming for view-invariant policies. We propose MoVie, a simple yet effective method for adapting visual **Mo**del-based policies to generalize to unseen **Vie**ws. MoVie leverages collected transitions from interactions and incorporates spatial transformer networks (STN [13]) in shallow layers, using the learning objective of the dynamics model (DM). Notably, MoVie requires no modifications during training and is compatible with various visual model-based RL algorithms. It only necessitates small-scale interactions for adaptation to the deployment view.

We perform extensive experiments on 7 robotic manipulation tasks (Adroit hand [20] and xArm [7]) and 11 locomotion tasks (DMControl suite [30])), across the proposed 4 view generalization settings, totaling $18 \times 4$ configurations. MoVie improves the view generalization ability substantially, compared to strong baselines including the inverse dynamics model (IDM [7]) and the dynamics model (DM). Remarkably, MoVie attains a relative improvement of $86\%$ in xArm and $152\%$ in Adroit, underscoring the potential of our method in robotics. **We are committed to releasing our code and testing platforms.** To conclude, our contributions are three-fold:

- We formulate the problem of view generalization in visual reinforcement learning with a wide range of tasks and settings that mimic real-world scenarios.
- We propose a simple model-based policy adaptation method for view generalization (**MoVie**), which incorporates STN into the shallow layers of the visual representation with a self-supervised dynamics prediction objective.
- We successfully showcase the effectiveness of our method through extensive experiments. The results serve as a testament to its capability and underscore its potential for practical deployment in robotic systems, particularly with complex camera views.

## 2   Related Work

**Visual generalization in reinforcement learning.** Agents trained by reinforcement learning (RL) from visual observations are prone to overfitting the training scenes, making it hard to generalize to

unseen environments with appearance differences. A large corpus of recent works has focused on addressing this issue [7, 9, 10, 14, 15, 19, 21, 31, 33, 35, 36]. Notably, SODA [10] provides a visual generalization benchmark to better evaluate the generalizability of policies, while they only consider appearance changes of agents and backgrounds. Distracting control suite [26] adds both appearance changes and camera view changes into DMControl [30], where the task diversity is limited.

**View generalization in robotics.** The field of robot learning has long grappled with the challenge of training models on limited views and achieving generalization to unseen views. Previous studies, such as RoboNet [3], have collected extensive video data encompassing various manipulation tasks. However, even with pre-training on such large-scale datasets, success rates on unseen views have only reached approximately $10\% \sim 20\%$ [3]. In recent efforts to tackle this challenge, researchers have primarily focused on third-person imitation learning [23–25] and view-invariant visual representations [2, 4, 16, 32, 37], but these approaches are constrained by the number of available camera views. In contrast, our work addresses a more demanding scenario where agents trained on a single fixed view are expected to generalize to diverse unseen views and dynamic camera settings.

**Test-time training.** There is a line of works that train neural networks at test-time with self-supervised learning in computer vision [5, 28, 29], robotics [27], and visual RL [7]. Specifically, PAD is the closest to our work [7], which adds an inverse dynamics model (IDM) objective into model-free policies for both training time and test time and gains better appearance generalization. In contrast, we differ in a lot of aspects: *(i)* we focus on visual model-based policies, *(ii)* we require no modification in training time, and *(iii)* our method is designed for view generalization specifically.

## 3 Preliminaries

**Formulation.** We model the problem as a Partially Observable Markov Decision Process (POMDP) $\mathcal{M} = \langle \mathcal{O}, \mathcal{A}, \mathcal{T}, \mathcal{R}, \gamma \rangle$, where $\mathbf{o} \in \mathcal{O}$ are observations, $\mathbf{a} \in \mathcal{A}$ are actions, $\mathcal{F} : \mathcal{O} \times \mathcal{A} \mapsto \mathcal{O}$ is a transition function (called dynamics as well), $r \in \mathcal{R}$ are rewards, and $\gamma \in [0, 1)$ is a discount factor. During training time, the agent's goal is to learn a policy $\pi$ that maximizes discounted cumulative rewards on $\mathcal{M}$, *i.e.*, $\max \mathbb{E}_\pi \left[ \sum_{t=0}^{\infty} \gamma^t r_t \right]$. During test time, the reward signal from the environment is not accessible to agents and only observations are available, which are possible to experience subtle changes such as appearance changes and camera view changes.

**Model-based reinforcement learning. TD-MPC** [11] is a model-based RL algorithm that combines model predictive control and temporal difference learning. TD-MPC learns a visual representation $\mathbf{z} = h(\mathbf{o})$ that maps the high-dimensional observation $\mathbf{o} \in \mathcal{O}$ into a latent state $\mathbf{z} \in \mathcal{Z}$ and a latent dynamics model $d : \mathcal{Z} \times \mathcal{A} \mapsto \mathcal{Z}$ that predicts the future latent state $\mathbf{z}' = d(\mathbf{z}, \mathbf{a})$ based on the current latent state $\mathbf{z}$ and the action $\mathbf{a}$. **MoDem** [8] accelerates TD-MPC with efficient utilization of expert demonstrations $\mathcal{D} = \{D_1, D_2, \cdots, D_N\}$ to solve challenging tasks such as dexterous manipulation [20]. In this work, we select TD-MPC and MoDem as the backbone algorithm to train model-based agents, while our algorithm could be easily extended to most model-based RL algorithms such as Dreamer [6].

## 4 Method

We propose a simple yet effective method, visual **Mo**del-based policy adaptation for **Vie**w generalization (**MoVie**), which can accommodate visual RL agents to novel camera views at test time.

**Learning objective for the test time.** Given a tuple $(\mathbf{o}_t, \mathbf{a}_t, \mathbf{o}_{t+1})$, the original latent state dynamics prediction objective can be written as

$$\mathcal{L}_{\text{dynamics}} = \|d(h(\mathbf{o_t}), \mathbf{a}_t) - h(\mathbf{o_{t+1}})\|_2, \qquad (1)$$

where $h$ is an image encoder that projects a high-dimensional observation from space $\mathcal{O}$ into a latent space $\mathcal{Z}$ and $d$ is a latent dynamics model $d : \mathcal{Z} \times \mathcal{A} \mapsto \mathcal{Z}$.

In test time, the observations under unseen views lie in a different space $\mathcal{O}'$, so that their corresponding latent space also changes to $\mathcal{Z}'$. However, the projection $h$ learned in training time can only map $\mathcal{O} \mapsto \mathcal{Z}$ while the policy $\pi$ only learns the mapping $\mathcal{Z} \mapsto \mathcal{A}$, thus making the policy hard to generalize to the correct mapping function $\mathcal{Z}' \mapsto \mathcal{A}$. Our proposal is thus to adapt the projection $h$ from a mapping function $h : \mathcal{O} \mapsto \mathcal{Z}$ to a more useful mapping function $h' : \mathcal{O}' \mapsto \mathcal{Z}$ so that the

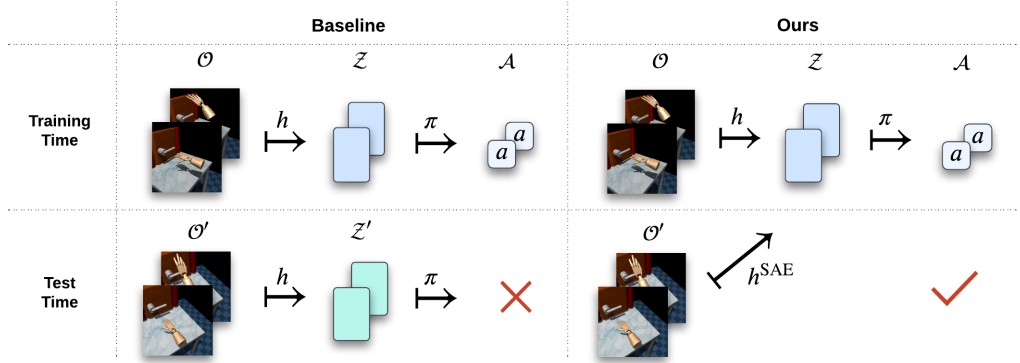

Figure 3: **An illustration of the reason why MoVie is effective.** We treat the frozen dynamics model as the source of supervision to adapt the latent space of $h^{\text{SAE}}$ to that of the training views.

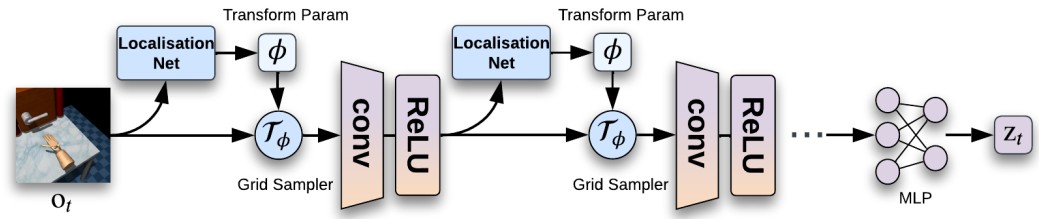

Figure 4: **The architecture of our spatial adaptive encoder (SAE).** We incorporate STNs before the first two convolutional layers of the encoder for better adaptation of the representation $h$.

policy would execute the correct mapping $\mathcal{Z} \mapsto \mathcal{A}$ without training. A vivid illustration is provided in Figure 3.

We freeze the latent dynamics model $d$, denoted as $d^\star$, so that the latent dynamics model is not a training target but a supervision. We also insert STN blocks [13] into the shallow layers of $h$ to better adapt the projection $h$, so that we write $h$ as $h^{\text{SAE}}$ (SAE denotes spatial adaptive encoder). Though the objective is still the latent state dynamics prediction loss, the supervision here is **superficially identical but fundamentally different** from training time. The formal objective is written as

$$\mathcal{L}_{\text{view}} = \|d^\star(h^{\text{SAE}}(\mathbf{o}), \mathbf{a}) - h^{\text{SAE}}(\mathbf{o_{t+1}})\|_2. \tag{2}$$

**Spatial adaptive encoder.** We now describe more details about our modified encoder architecture during test time, referred to as spatial adaptive encoder (SAE). To keep our method simple and fast to adapt, we only insert two different STNs into the original encoder, as shown in Figure 4. We observe in our experiments that transforming the low-level features (*i.e.*, RGB features and shallow layer features) is most critical for adaptation, while the benefit of adding more STNs is limited (see Table 6). An STN block consists of two parts: *(i)* a localisation net that predicts an affine transformation with 6 parameters and *(ii)* a grid sampler that generates an affined grid and samples features from the original feature map. The point-wise affine transformation is written as

$$\begin{pmatrix} x^s \\ y^s \end{pmatrix} = \mathcal{T}_\phi(G) = \mathrm{A}_\phi \begin{pmatrix} x^t \\ y^t \\ 1 \end{pmatrix} = \begin{bmatrix} \phi_{11} & \phi_{12} & \phi_{13} \\ \phi_{21} & \phi_{22} & \phi_{23} \end{bmatrix} \begin{pmatrix} x^t \\ y^t \\ 1 \end{pmatrix} \tag{3}$$

where $G$ is the sampling grid, $(x_i^t, y_i^t)$ are the target coordinates of the regular grid in the output feature map, $(x_i^s, y_i^s)$ are the source coordinates in the input feature map that define the sample points, and $\mathrm{A}_\phi$ is the affine transformation matrix.

**Training strategy for SAE.** We use a learning rate $1 \times 10^{-5}$ for STN layers and $1 \times 10^{-7}$ for the encoder. We utilize a replay buffer with size 256 to store history observations and update 32 times for each time step to heavily utilize the online data. Implementation details remain in Appendix A.

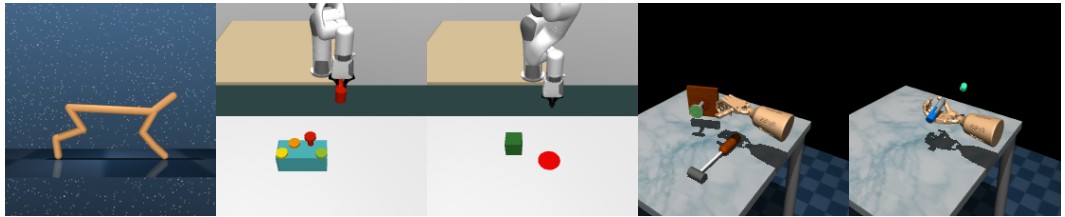

Figure 5: **Visualization of three task domains**, including DMControl [30], xArm [7], Adroit [20].

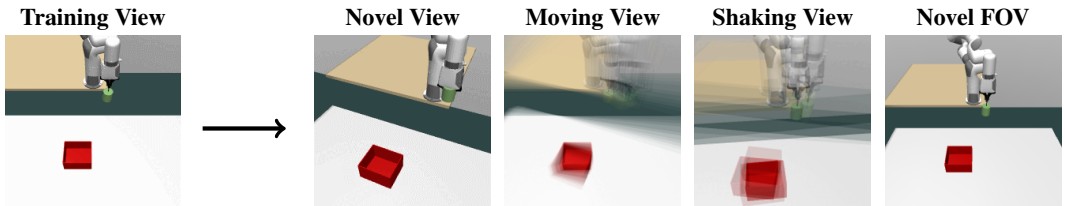

| Training View | Novel View | Moving View | Shaking View | Novel FOV |

Figure 6: **Visualization of the training view and four view generalization settings.** Trajectory visualization for all tasks is available on our website yangsizhe.github.io/MoVie.

## 5 Experiments

In this section, we investigate how well an agent trained on a single fixed view generalizes to unseen views during test time. During the evaluation, agents have no access to reward signals, presenting a significant challenge for agents to self-supervise using online data.

### 5.1 Experiment Setup

**Formulation of camera view variations**: *a)* **novel view**, where we maintain a fixed camera target while adjusting the camera position in both horizontal and vertical directions by a certain margin, *b)* **moving view**, where we establish a predefined trajectory encompassing the scene and the camera follows this trajectory, moving back and forth for each time step while focusing on the center of the scene, *c)* **shaking view**, where we add Gaussian noise onto the original camera position at each time step, and *d)* **novel FOV**, where the FOV of the camera is altered once, different from the training phase. A visualization of four settings is provided in Figure 6 and videos are available in yangsizhe.github.io/MoVie for a better understanding of our configurations. Details remain in Appendix B.

**Tasks.** Our test platform consists of **18** tasks from **3** domains: 11 tasks from DMControl [30], 3 tasks from Adroit [20], and 4 tasks from xArm [7]. A visualization of the three domains is in Figure 5. Although the primary motivation for this study is for addressing view generalization in real-world robot learning, which has not yet been conducted, we contend that the extensive range of tasks tackled in our research effectively illustrates the potential of MoVie for real-world application. We run 3 seeds per experiment with seed numbers $0, 1, 2$ and run 20 episodes per seed. During these 20 episodes, the models could store the history transitions. We report cumulative rewards for DMControl tasks and success rates for robotic manipulation tasks, averaging over episodes.

**Baselines.** We first train visual model-based policies with TD-MPC [11] on DMControl and xArm environments and MoDem [8] on Adroit environments under the default configurations and then test these policies in the view generalization setting. Since MoVie consists of two components mainly (*i.e.*, DM and STN), we build two test-time adaptation algorithms by replacing each module: *a)* **DM**, which removes STN from MoVie and *b)* **IDM+STN**, which replaces DM in MoVie with IDM [7]. IDM+STN is very close to PAD [7], and we add STN for fair comparison. We keep all training settings the same.

Table 1: **Experiment conclusion across all domains and all settings.** The best method on each setting is in **bold** and the relative improvement over TD-MPC is also reported.

| Setting | Domain | TD-MPC | DM | IDM+STN | MoVie |
|---|---|---|---|---|---|
| Novel view | DMControl | 395.61 | 508.85 (↑29%) | 378.80 (↓4%) | **623.19** (↑**57%**) |
| | xArm | 16 | 36 (↑125%) | 18 (↑13%) | **46** (↑**188%**) |
| | Adroit | 8 | 14 (↑75%) | 12 (↑50%) | **34** (↑**325%**) |
| Moving view | DMControl | 605.98 | 611.87 (↑1%) | 502.34 (↓17%) | **673.74** (↑**11%**) |
| | xArm | 20 | 31 (↑55%) | 24 (↑20%) | **42** (↑**110%**) |
| | Adroit | 15 | 20 (↑33%) | 15 (↑0%) | **45** (↑**200%**) |
| Shaking view | DMControl | 441.79 | 348.26 (↓21%) | 291.45 (↓26%) | **558.23** (↑**26%**) |
| | xArm | 42 | 44 (↑5%) | 44 (↑5%) | **45** (↑**7%**) |
| | Adroit | 30 | 35 (↑17%) | 26 (↓13%) | **63** (↑**110%**) |
| Novel FOV | DMControl | 527.47 | 613.74 (↑16%) | 542.43 (↑3%) | **770.56** (↑**46%**) |
| | xArm | 34 | 47 (↑38%) | 40 (↑18%) | **75** (↑**121%**) |
| | Adroit | 31 | 51 (↑65%) | 31 (↑0%) | **68** (↑**119%**) |
| **All settings** | DMControl | 492.71 | 520.68 (↑6%) | 426.76 (↓13%) | **656.43** (↑**33%**) |
| | xArm | 28 | 40 (↑43%) | 32 (↑14%) | **52** (↑**86%**) |
| | Adroit | 21 | 30 (↑43%) | 21 (↑0%) | **53** (↑**152%**) |

Table 2: **Results in novel view.** The best method on each task is in **bold**.

| Novel view | TD-MPC | DM | IDM+STN | MoVie |
|---|---|---|---|---|
| Cheetah, run | $90.13_{\pm20.38}$ | $254.43_{\pm13.89}$ | $127.76_{\pm16.16}$ | $\mathbf{342.39_{\pm54.95}}$ |
| Walker, walk | $249.34_{\pm14.78}$ | $262.65_{\pm62.86}$ | $215.90_{\pm29.30}$ | $\mathbf{512.71_{\pm404.25}}$ |
| Walker, stand | $568.01_{\pm16.81}$ | $635.64_{\pm13.70}$ | $508.82_{\pm15.77}$ | $\mathbf{679.90_{\pm23.03}}$ |
| Walker, run | $127.07_{\pm8.06}$ | $\mathbf{131.65_{\pm2.20}}$ | $94.49_{\pm18.72}$ | $94.49_{\pm18.87}$ |
| Cup, catch | $922.83_{\pm30.74}$ | $949.36_{\pm9.20}$ | $932.75_{\pm23.34}$ | $\mathbf{961.98_{\pm2.68}}$ |
| Finger, spin | $137.30_{\pm4.47}$ | $750.65_{\pm9.90}$ | $141.03_{\pm37.52}$ | $\mathbf{892.01_{\pm1.20}}$ |
| Finger, turn-easy | $380.80_{\pm138.54}$ | $528.91_{\pm160.25}$ | $348.00_{\pm123.84}$ | $\mathbf{705.36_{\pm71.98}}$ |
| Finger, turn-hard | $261.43_{\pm72.83}$ | $312.36_{\pm134.28}$ | $306.76_{\pm181.13}$ | $\mathbf{331.83_{\pm18.98}}$ |
| Pendulum, swingup | $40.65_{\pm3.72}$ | $64.51_{\pm2.17}$ | $82.46_{\pm8.19}$ | $\mathbf{528.76_{\pm77.51}}$ |
| Reacher, easy | $981.25_{\pm4.80}$ | $982.30_{\pm0.83}$ | $875.31_{\pm150.91}$ | $\mathbf{984.66_{\pm2.78}}$ |
| Reacher, hard | $592.91_{\pm59.96}$ | $724.91_{\pm125.84}$ | $533.50_{\pm96.90}$ | $\mathbf{821.03_{\pm67.89}}$ |
| DMControl | 395.61 | 508.85 | 378.80 | **623.19** (↑**57%**) |
| xArm, reach | $3_{\pm2}$ | $85_{\pm13}$ | $1_{\pm2}$ | $\mathbf{95_{\pm0}}$ |
| xArm, push | $46_{\pm5}$ | $45_{\pm8}$ | $\mathbf{60_{\pm5}}$ | $48_{\pm7}$ |
| xArm, peg in box | $5_{\pm0}$ | $5_{\pm15}$ | $3_{\pm2}$ | $\mathbf{6_{\pm7}}$ |
| xArm, hammer | $10_{\pm10}$ | $10_{\pm10}$ | $8_{\pm5}$ | $\mathbf{36_{\pm12}}$ |
| xArm | 16 | 36 | 18 | **46** (↑**188%**) |
| Adroit, door | $0_{\pm0}$ | $10_{\pm5}$ | $0_{\pm0}$ | $\mathbf{66_{\pm7}}$ |
| Adroit, hammer | $6_{\pm2}$ | $11_{\pm10}$ | $\mathbf{21_{\pm2}}$ | $11_{\pm5}$ |
| Adroit, pen | $18_{\pm5}$ | $20_{\pm8}$ | $15_{\pm5}$ | $\mathbf{25_{\pm18}}$ |
| Adroit | 8 | 14 | 12 | **34** (↑**325%**) |

## 5.2 Main Experiment Results

Considering the extensive scale of our conducted experiments, we present an overview of our findings in Table 1 and Figure 1. The detailed results for four configurations are provided in Table 2, Table 3, Table 4, and Table 5 respectively. We then detail our findings below.

**Superiority of MoVie across domains, especially in robotic manipulation.** It is observed that irrespective of the domain or configuration, MoVie consistently outperforms TD-MPC without any adaptation and other methods that apply DM and IDM. This large improvement highlights the fundamental role of our straightforward approach in addressing view generalization. Additionally, we observe that MoVie exhibits greater suitability for robotic manipulation tasks. We observe a significant relative improvement of **86**% in xArm tasks and **152**% in Adroit tasks, as opposed to a comparatively modest improvement of only 33% in DMControl tasks; see Table 1. We attribute this

Table 3: **Results in moving view.** The best method on each task is in **bold**.

| Moving view | TD-MPC | DM | IDM+STN | MoVie |
|---|---|---|---|---|
| Cheetah, run | $235.37_{\pm63.39}$ | $344.70_{\pm7.54}$ | $199.48_{\pm25.78}$ | $\mathbf{365.22_{\pm42.66}}$ |
| Walker, walk | $632.77_{\pm15.62}$ | $707.60_{\pm26.72}$ | $373.44_{\pm30.85}$ | $\mathbf{810.19_{\pm7.77}}$ |
| Walker, stand | $\mathbf{803.97_{\pm15.75}}$ | $706.05_{\pm5.75}$ | $576.87_{\pm44.69}$ | $712.48_{\pm11.67}$ |
| Walker, run | $\mathbf{295.54_{\pm4.39}}$ | $250.84_{\pm7.02}$ | $190.02_{\pm2.61}$ | $281.43_{\pm33.34}$ |
| Cup, catch | $887.20_{\pm3.67}$ | $910.51_{\pm4.12}$ | $915.16_{\pm7.31}$ | $\mathbf{951.26_{\pm10.68}}$ |
| Finger, spin | $636.91_{\pm3.91}$ | $697.11_{\pm8.67}$ | $532.90_{\pm3.25}$ | $\mathbf{896.00_{\pm21.65}}$ |
| Finger, turn-easy | $715.45_{\pm19.23}$ | $728.93_{\pm180.26}$ | $683.85_{\pm112.64}$ | $\mathbf{744.45_{\pm16.54}}$ |
| Finger, turn-hard | $\mathbf{593.46_{\pm48.22}}$ | $454.58_{\pm88.50}$ | $559.20_{\pm121.35}$ | $558.26_{\pm25.64}$ |
| Pendulum, swingup | $26.23_{\pm1.85}$ | $83.88_{\pm10.55}$ | $47.20_{\pm4.39}$ | $\mathbf{236.81_{\pm50.23}}$ |
| Reacher, easy | $\mathbf{984.10_{\pm4.38}}$ | $981.78_{\pm0.60}$ | $695.03_{\pm236.91}$ | $982.58_{\pm1.39}$ |
| Reacher, hard | $854.76_{\pm31.64}$ | $864.55_{\pm556.37}$ | $752.55_{\pm83.55}$ | $\mathbf{872.46_{\pm51.30}}$ |
| DMControl | 605.98 | 611.87 | 502.34 | **673.74** (↑**11%**) |
| xArm, reach | $15_{\pm5}$ | $71_{\pm2}$ | $21_{\pm2}$ | $\mathbf{80_{\pm21}}$ |
| xArm, push | $58_{\pm5}$ | $52_{\pm10}$ | $55_{\pm5}$ | $\mathbf{73_{\pm5}}$ |
| xArm, peg in box | $0_{\pm0}$ | $0_{\pm0}$ | $1_{\pm2}$ | $\mathbf{5_{\pm8}}$ |
| xArm, hammer | $8_{\pm7}$ | $0_{\pm0}$ | $\mathbf{10_{\pm5}}$ | $8_{\pm7}$ |
| xArm | 20 | 31 | 24 | **42** (↑**110%**) |
| Adroit, door | $0_{\pm0}$ | $10_{\pm5}$ | $0_{\pm0}$ | $\mathbf{66_{\pm7}}$ |
| Adroit, hammer | $25_{\pm8}$ | $31_{\pm12}$ | $28_{\pm02}$ | $\mathbf{43_{\pm18}}$ |
| Adroit, pen | $20_{\pm0}$ | $20_{\pm10}$ | $18_{\pm14}$ | $\mathbf{26_{\pm2}}$ |
| Adroit | 15 | 20 | 15 | **45** (↑**200%**) |

Table 4: **Results in shaking view.** The best method on each task is in **bold**.

| Shaking view | TD-MPC | DM | IDM+STN | MoVie |
|---|---|---|---|---|
| Cheetah, run | $381.04_{\pm39.31}$ | $317.66_{\pm9.57}$ | $212.31_{\pm19.74}$ | $\mathbf{493.54_{\pm56.80}}$ |
| Walker, walk | $662.13_{\pm57.71}$ | $627.77_{\pm21.47}$ | $340.48_{\pm36.92}$ | $\mathbf{835.99_{\pm14.98}}$ |
| Walker, stand | $\mathbf{834.70_{\pm1.71}}$ | $604.55_{\pm16.20}$ | $471.30_{\pm69.19}$ | $687.96_{\pm9.11}$ |
| Walker, run | $251.58_{\pm13.69}$ | $186.83_{\pm3.75}$ | $128.31_{\pm9.19}$ | $\mathbf{291.39_{\pm7.94}}$ |
| Cup, catch | $752.86_{\pm41.81}$ | $835.16_{\pm32.91}$ | $648.75_{\pm12.77}$ | $\mathbf{951.20_{\pm21.01}}$ |
| Finger, spin | $89.55_{\pm4.72}$ | $88.56_{\pm5.86}$ | $145.68_{\pm2.93}$ | $\mathbf{284.05_{\pm473.73}}$ |
| Finger, turn-easy | $\mathbf{717.60_{\pm40.93}}$ | $449.10_{\pm20.48}$ | $656.23_{\pm50.35}$ | $694.20_{\pm91.02}$ |
| Finger, turn-hard | $411.41_{\pm40.28}$ | $383.96_{\pm32.94}$ | $132.40_{\pm34.25}$ | $\mathbf{629.20_{\pm110.57}}$ |
| Pendulum, swingup | $23.73_{\pm9.24}$ | $\mathbf{57.41_{\pm4.50}}$ | $27.30_{\pm2.38}$ | $48.23_{\pm30.25}$ |
| Reacher, easy | $529.58_{\pm65.05}$ | $225.08_{\pm31.02}$ | $299.11_{\pm59.00}$ | $\mathbf{771.38_{\pm150.60}}$ |
| Reacher, hard | $205.53_{\pm25.16}$ | $54.80_{\pm32.45}$ | $144.11_{\pm5.60}$ | $\mathbf{453.48_{\pm381.60}}$ |
| DMControl | 441.79 | 348.26 | 291.45 | **558.23** (↑**26%**) |
| xArm, reach | $76_{\pm2}$ | $76_{\pm7}$ | $83_{\pm2}$ | $\mathbf{88_{\pm20}}$ |
| xArm, push | $\mathbf{64_{\pm3}}$ | $55_{\pm10}$ | $61_{\pm5}$ | $57_{\pm13}$ |
| xArm, peg in box | $3_{\pm2}$ | $\mathbf{15_{\pm13}}$ | $3_{\pm2}$ | $5_{\pm5}$ |
| xArm, hammer | $25_{\pm5}$ | $28_{\pm7}$ | $28_{\pm12}$ | $\mathbf{30_{\pm13}}$ |
| xArm | 42 | 44 | 44 | **45** (↑**7%**) |
| Adroit, door | $1_{\pm2}$ | $10_{\pm0}$ | $5_{\pm5}$ | $\mathbf{83_{\pm2}}$ |
| Adroit, hammer | $58_{\pm12}$ | $65_{\pm18}$ | $55_{\pm20}$ | $\mathbf{70_{\pm27}}$ |
| Adroit, pen | $30_{\pm5}$ | $30_{\pm8}$ | $18_{\pm5}$ | $\mathbf{35_{\pm0}}$ |
| Adroit | 30 | 35 | 26 | **63** (↑**110%**) |

disparity to two factors: *a)* the inherent complexity of robotic manipulation tasks and *b)* the pressing need for effective view generalization in this domain.

**Challenges in handling shaking view.** Despite improvements of MoVie in various settings, we have identified a relative weakness in addressing the *shaking view* scenario. For instance, in xArm tasks, the success rate of MoVie is only 45%, which is close to the 42% success rate of TD-MPC without adaptation. Other baselines such as DM and IDM+STN also experience performance drops. We acknowledge the inherent difficulty of the shaking view scenario, while it is worth noting that in real-world robotic manipulation applications, cameras often exhibit smoother movements or are positioned in fixed views, partially mitigating the impact of shaking view.

Table 5: **Novel FOV.** The best method on each task is in **bold**.

| Novel FOV | TD-MPC | DM | IDM+STN | MoVie |
|---|---|---|---|---|
| Cheetah, run | $128.55_{\pm6.57}$ | $379.01_{\pm10.90}$ | $299.02_{\pm88.47}$ | $\mathbf{532.94_{\pm19.74}}$ |
| Walker, walk | $239.54_{\pm129.47}$ | $882.21_{\pm12.75}$ | $579.58_{\pm47.87}$ | $\mathbf{920.18_{\pm8.76}}$ |
| Walker, stand | $679.70_{\pm21.21}$ | $732.07_{\pm14.77}$ | $678.46_{\pm55.32}$ | $\mathbf{785.52_{\pm24.45}}$ |
| Walker, run | $184.28_{\pm1.39}$ | $337.28_{\pm11.40}$ | $295.99_{\pm10.00}$ | $\mathbf{482.03_{\pm13.84}}$ |
| Cup, catch | $940.20_{\pm28.21}$ | $912.70_{\pm34.54}$ | $964.83_{\pm10.45}$ | $\mathbf{973.60_{\pm1.94}}$ |
| Finger, spin | $675.81_{\pm4.99}$ | $886.63_{\pm4.54}$ | $414.83_{\pm79.20}$ | $\mathbf{917.21_{\pm1.71}}$ |
| Finger, turn-easy | $792.26_{\pm145.24}$ | $690.61_{\pm140.88}$ | $805.63_{\pm54.95}$ | $\mathbf{845.35_{\pm35.52}}$ |
| Finger, turn-hard | $591.43_{\pm109.84}$ | $393.18_{\pm118.38}$ | $449.88_{\pm121.99}$ | $\mathbf{640.21_{\pm183.23}}$ |
| Pendulum, swingup | $216.90_{\pm78.56}$ | $184.33_{\pm83.99}$ | $253.36_{\pm78.26}$ | $\mathbf{699.90_{\pm50.64}}$ |
| Reacher, easy | $929.40_{\pm12.35}$ | $931.56_{\pm12.69}$ | $937.20_{\pm45.97}$ | $\mathbf{971.48_{\pm11.07}}$ |
| Reacher, hard | $424.11_{\pm83.74}$ | $421.56_{\pm94.93}$ | $287.96_{\pm6.33}$ | $\mathbf{707.75_{\pm86.05}}$ |
| `DMControl` | 527.47 | 613.74 | 542.43 | $\mathbf{770.56}$ (↑**46%**) |
| xArm, reach | $53_{\pm12}$ | $98_{\pm2}$ | $85_{\pm5}$ | $\mathbf{100_{\pm0}}$ |
| xArm, push | $60_{\pm13}$ | $71_{\pm5}$ | $60_{\pm5}$ | $\mathbf{81_{\pm2}}$ |
| xArm, peg in box | $20_{\pm5}$ | $13_{\pm10}$ | $6_{\pm11}$ | $\mathbf{63_{\pm11}}$ |
| xArm, hammer | $3_{\pm2}$ | $6_{\pm5}$ | $8_{\pm2}$ | $\mathbf{55_{\pm5}}$ |
| `xArm` | 34 | 47 | 40 | $\mathbf{75}$ (↑**121%**) |
| Adroit, door | $1_{\pm2}$ | $58_{\pm10}$ | $3_{\pm5}$ | $\mathbf{81_{\pm12}}$ |
| Adroit, hammer | $66_{\pm7}$ | $75_{\pm18}$ | $76_{\pm10}$ | $\mathbf{81_{\pm10}}$ |
| Adroit, pen | $26_{\pm10}$ | $20_{\pm8}$ | $15_{\pm5}$ | $\mathbf{41_{\pm7}}$ |
| `Adroit` | 31 | 51 | 31 | $\mathbf{68}$ (↑**119%**) |

**Effective adaptation in novel view, moving view, and novel FOV scenarios.** In addition to the shaking view setting, MoVie consistently outperforms TD-MPC without adaptation by $2\times \sim 4\times$ in robotic manipulation tasks across the other three settings. It is worth noting that these three settings are among the most common scenarios encountered in real-world applications.

**Real-world implications.** Our findings have important implications for real-world deployment of robots. Previous methods, relying on domain randomization and extensive data augmentation during training, often hinder the learning process. Our proposed method enables direct deployment of offline or simulation-trained agents, improving success rates with minimal interactions.

## 5.3 Ablations

To validate the rationale behind several choices of MoVie and our test platform, we perform a comprehensive set of ablation experiments.

**Integration of STN with low-level features.** To enhance view generalization, we incorporate two STN blocks [13], following the image observation and the initial convolutional layer of the feature encoder. This integration is intended to align the low-level features with the training view, thereby preserving the similarity of deep semantic features to the training view. As shown in Table 6, by progressively adding more layers to the feature encoder, we observe that deeper layers do not provide significant additional benefits, supporting our intuition for view generalization.

Table 6: **Ablation on applying different numbers of STNs from shallow to deep.** The best method on each setting is in **bold**.

| Cheetah-run | 0 | 1 | 2 (MoVie) | 3 | 4 |
|---|---|---|---|---|---|
| Novel View | $254.42_{\pm13.89}$ | $259.64_{\pm32.01}$ | $\mathbf{342.39_{\pm54.95}}$ | $309.25_{\pm33.19}$ | $327.39_{\pm10.44}$ |
| Moving View | $344.70_{\pm7.54}$ | $360.29_{\pm13.64}$ | $\mathbf{365.22_{\pm42.66}}$ | $361.28_{\pm6.78}$ | $357.29_{\pm31.24}$ |
| Shaking View | $317.66_{\pm9.57}$ | $491.96_{\pm20.07}$ | $\mathbf{493.54_{\pm56.80}}$ | $454.82_{\pm19.57}$ | $472.57_{\pm8.49}$ |
| Novel FOV | $379.01_{\pm10.90}$ | $512.69_{\pm17.67}$ | $\mathbf{532.94_{\pm19.74}}$ | $505.65_{\pm12.02}$ | $492.79_{\pm21.66}$ |

**Different novel views.** We classify novel views into three levels of difficulty, with our main experiments employing the *medium* difficulty level by default. Table 7 presents additional results for

the *easy* and *hard* difficulty levels. As the difficulty level increases, we observe a consistent decrease in the performance of all the methods.

Table 7: **Ablation on different novel views.** The best method on each setting is in **bold**.

| Cheetah-run | TD-MPC | DM | IDM+STN | MoVie |
|---|---|---|---|---|
| Easy | $265.90_{\pm 7.94}$ | $441.61_{\pm 14.00}$ | $349.86_{\pm 55.82}$ | $\mathbf{466.67_{\pm 58.94}}$ |
| Medium | $90.13_{\pm 20.38}$ | $254.43_{\pm 13.89}$ | $127.76_{\pm 16.16}$ | $\mathbf{342.39_{\pm 54.95}}$ |
| Hard | $48.64_{\pm 13.20}$ | $112.39_{\pm 8.36}$ | $59.64_{\pm 5.35}$ | $\mathbf{215.36_{\pm 62.03}}$ |

Table 8: **Ablation on IDM with and without STN.** The best method is in **bold**.

| Task | Setting | IDM | IDM+STN |
|---|---|---|---|
| xArm, push | Novel view | $40_{\pm 5}$ | $\mathbf{60_{\pm 5}}$ |
| | Moving view | $46_{\pm 2}$ | $\mathbf{55_{\pm 5}}$ |
| | Shaking view | $60_{\pm 10}$ | $\mathbf{61_{\pm 5}}$ |
| | Novel FOV | $58_{\pm 2}$ | $\mathbf{60_{\pm 5}}$ |
| | All settings | $51$ | $\mathbf{59}$ |
| xArm, hammer | Novel view | $6_{\pm 5}$ | $\mathbf{8_{\pm 5}}$ |
| | Moving view | $5_{\pm 0}$ | $\mathbf{10_{\pm 5}}$ |
| | Shaking view | $\mathbf{35_{\pm 8}}$ | $28_{\pm 12}$ |
| | Novel FOV | $6_{\pm 7}$ | $\mathbf{8_{\pm 2}}$ |
| | All settings | $13$ | $\mathbf{14}$ |
| Cup, catch | Novel view | $867.87_{\pm 3.14}$ | $\mathbf{932.75_{\pm 23.34}}$ |
| | Moving view | $912.85_{\pm 10.22}$ | $\mathbf{915.16_{\pm 7.31}}$ |
| | Shaking view | $435.40_{\pm 87.32}$ | $\mathbf{648.75_{\pm 12.77}}$ |
| | Novel FOV | $815.27_{\pm 34.61}$ | $\mathbf{964.83_{\pm 10.45}}$ |
| | All settings | $757.84$ | $\mathbf{865.37}$ |
| Finger, spin | Novel view | $\mathbf{177.26_{\pm 2.49}}$ | $141.03_{\pm 37.52}$ |
| | Moving view | $332.90_{\pm 0.28}$ | $\mathbf{532.90_{\pm 3.25}}$ |
| | Shaking view | $106.72_{\pm 7.11}$ | $\mathbf{145.68_{\pm 2.93}}$ |
| | Novel FOV | $311.66_{\pm 8.55}$ | $\mathbf{414.83_{\pm 79.20}}$ |
| | All settings | $232.13$ | $\mathbf{301.86}$ |
| Cheetah, run | Novel view | $111.73_{\pm 18.98}$ | $\mathbf{127.76_{\pm 16.16}}$ |
| | Moving view | $\mathbf{205.19_{\pm 25.45}}$ | $199.48_{\pm 25.78}$ |
| | Shaking view | $210.06_{\pm 27.99}$ | $\mathbf{212.32_{\pm 19.74}}$ |
| | Novel FOV | $262.76_{\pm 62.17}$ | $\mathbf{299.03_{\pm 88.47}}$ |
| | All settings | $197.43$ | $\mathbf{209.43}$ |

**The efficacy of STN in conjunction with IDM.** Curious readers might be concerned about our direct utilization of IDM+STN in the main experiments, suggesting that STN could potentially be detrimental to IDM. However, Table 8 shows that STN not only benefits our method but also improves the performance of IDM, thereby demonstrating effectiveness of SAE for adaptation of the representation and validating our baseline selection.

**Finetune or freeze the DM.** In our approach, we employ the frozen DM as a form of supervision to guide the adaptation process of the encoder. However, it remains unverified for the readers whether end-to-end finetuning of both the DM and the encoder would yield similar benefits. The results presented in Table 9 demonstrate that simplistic end-to-end finetuning does not outperform MoVie, thereby reinforcing the positive results obtained by MoVie.

Table 9: **Ablation on whether to finetune the DM.** The best method is in **bold**.

| Task | Setting | Finetune DM | MoVie |
|---|---|---|---|
| xArm, push | Novel view | $\mathbf{51}_{\pm\mathbf{2}}$ | $48_{\pm7}$ |
| | Moving view | $48_{\pm7}$ | $\mathbf{73}_{\pm\mathbf{5}}$ |
| | Shaking view | $45_{\pm5}$ | $\mathbf{57}_{\pm\mathbf{13}}$ |
| | Novel FOV | $68_{\pm5}$ | $\mathbf{81}_{\pm\mathbf{2}}$ |
| | All settings | 53 | **64** |
| xArm, hammer | Novel view | $10_{\pm5}$ | $\mathbf{36}_{\pm\mathbf{12}}$ |
| | Moving view | $0_{\pm0}$ | $\mathbf{8}_{\pm\mathbf{7}}$ |
| | Shaking view | $5_{\pm5}$ | $\mathbf{30}_{\pm\mathbf{13}}$ |
| | Novel FOV | $11_{\pm2}$ | $\mathbf{55}_{\pm\mathbf{5}}$ |
| | All settings | 6 | **32** |
| Cup, catch | Novel view | $648.90_{\pm25.668}$ | $\mathbf{961.98}_{\pm\mathbf{2.68}}$ |
| | Moving view | $676.05_{\pm79.69}$ | $\mathbf{951.26}_{\pm\mathbf{10.68}}$ |
| | Shaking view | $228.20_{\pm7.84}$ | $\mathbf{951.20}_{\pm\mathbf{21.01}}$ |
| | Novel FOV | $658.50_{\pm8.48}$ | $\mathbf{973.60}_{\pm\mathbf{1.94}}$ |
| | All settings | 552.91 | **959.51** |
| Finger, spin | Novel view | $278.57_{\pm26.34}$ | $\mathbf{892.01}_{\pm\mathbf{1.20}}$ |
| | Moving view | $192.95_{\pm72.76}$ | $\mathbf{896.00}_{\pm\mathbf{21.65}}$ |
| | Shaking view | $1.50_{\pm0.70}$ | $\mathbf{284.05}_{\pm\mathbf{473.73}}$ |
| | Novel FOV | $372.82_{\pm51.63}$ | $\mathbf{917.21}_{\pm\mathbf{1.71}}$ |
| | All settings | 211.46 | **747.31** |
| Cheetah, run | Novel view | $273.01_{\pm11.29}$ | $\mathbf{342.39}_{\pm\mathbf{54.95}}$ |
| | Moving view | $331.55_{\pm22.22}$ | $\mathbf{365.22}_{\pm\mathbf{42.66}}$ |
| | Shaking view | $476.25_{\pm30.25}$ | $\mathbf{493.54}_{\pm\mathbf{56.80}}$ |
| | Novel FOV | $\mathbf{561.94}_{\pm\mathbf{37.73}}$ | $532.94_{\pm19.74}$ |
| | All settings | 410.68 | **433.52** |

## 6   Conclusion

In this study, we present MoVie, a method for adapting visual model-based policies to achieve view generalization. MoVie mainly finetunes a spatial adaptive image encoder using the objective of the latent state dynamics model during test time. Notably, we maintain the dynamics model in a frozen state, allowing it to function as a form of self-supervision. Furthermore, we categorize the view generalization problem into four distinct settings: novel view, moving view, shaking view, and novel FOV. Through a systematic evaluation of MoVie on 18 tasks across these four settings, totaling 64 different configurations, we demonstrate its general and remarkable effectiveness.

One limitation of our work is the lack of real robot experiments, while our focus is on addressing view generalization in robot datasets and deploying visual reinforcement learning agents in real-world scenarios. In our future work, we would evaluate MoVie on real-world robotic tasks.

## Acknowledgment

This work is supported by National Key R&D Program of China (2022ZD0161700).

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

# Appendix

## A    Implementation Details

In this section, we describe the implementation details of our algorithm for training on the training view and test time training in view generalization settings on the DMControl [30], xArm [7], and Adroit [20] environments. We utilize the official implementation of TD-MPC [11] and MoDem [8] which are available at github.com/nicklashansen/tdmpc and github.com/facebookresearch/modem as the model-based reinforcement learning codebase. During training time, we use the default hyperparameters in official implementation of TD-MPC and MoDem. We present relevant hyperparameters during both training and test time in Table 10 and Table 11. One seed of our experiments could be run on a single 3090 GPU with fewer than 2GB and it takes $\sim 1$ hours for test-time training.

**Training time setup.** We train visual model-based policies with TD-MPC on DMControl and xArm environments, and MoDem on Adroit environments, We employ identical network architecture and hyperparameters as original TD-MPC and MoDem during training time.

The network architecture of the encoder in original TD-MPC is composed of a stack of 4 convolutional layers, each with 32 filters, no padding, stride of 2, $7 \times 7$ kernels for the first one, $5 \times 5$ kernels for the second one and $3 \times 3$ kernels for all others, yielding a final feature map of dimension $3 \times 3 \times 32$ (inputs whose framestack is 3 have dimension $84 \times 84 \times 9$). After the convolutional layers, a fully connected layer with an input size of 288 performs a linear transformation on the input and generates a 50-dimensional vector as the final output.

The network architecture of the encoder in original Modem is composed of a stack of 6 convolutional layers, each with 32 filters, no padding, stride of 2, $7 \times 7$ kernels for the first one, $5 \times 5$ kernels for the second one and $3 \times 3$ kernels for all others, yielding a final feature map of dimension $2 \times 2 \times 32$ (inputs whose framestack is 2 have dimension $224 \times 224 \times 6$). After the convolutional layers, a fully connected layer with an input size of 128 performs a linear transformation on the input and generates a 50-dimensional vector as the final output.

**Test time training setup.** During test time, we train spatial adaptive encoder (SAE) to adapt to view changes. We insert STN blocks before and after the first convolutional layer of the original encoders in TD-MPC and MoDem. The original encoders are augmented by inserting STN blocks, resulting in the formation of SAE. Particularly, for the STN block inserted before the first convolutional layer, the input is a single frame. This means that when the frame stack size is N, N individual frames are fed into this STN block. This is done to apply different transformations to different frames in cases of moving and shaking view.

To update the SAE, we collect online data using a buffer with a size of 256. For each update, we randomly sample 32 (observation, action, next_observation) tuples from the buffer as a batch. The optimization objective is to minimize the loss in predicting the dynamics of the latent states, as defined in Equation 2.

During testing on each task, we run 20 consecutive episodes, although typically only a few or even less than one episode is needed for the test-time training to converge. To make efficient use of the data collected with minimal interactions, we employ a multi-update strategy. After each interaction with the environment, the SAE is updated 32 times.

The following is the network architecture of the first STN block inserted into the encoder of TD-MPC.

```
STN_Block_0_TDMPC(
  (localization): Sequential(
    # By default, each image consists of three channels. Each frame in the
        ↪ observation is treated as an independent input to the STN.
    (0): Conv2d(in_channels=3, out_channels=8, kernel_size=7, stride=1)
    (1): MaxPool2d(kernel_size=4, stride=4, padding=0)
    (2): ReLU()
    (3): Conv2d(in_channels=8, out_channels=10, kernel_size=5, stride=1)
    (4): MaxPool2d(kernel_size=4, stride=4, padding=0)
    (5): ReLU()
  )
  (fc_loc): Sequential(
    (0): Linear(in_dim=90, out_dim=32)
```

```
      (1): ReLU()
      (2): Linear(in_dim=32, out_dim=6)
    )
  )
```

The following is the network architecture of the second STN block inserted into the encoder of TD-MPC.

```
STN_Block_1_TDMPC(
  (localization): Sequential(
    (0): Conv2d(in_channels=32, out_channels=8, kernel_size=7, stride=1)
    (1): MaxPool2d(kernel_size=3, stride=3, padding=0)
    (2): ReLU()
    (3): Conv2d(in_channels=8, out_channels=10, kernel_size=5, stride=1)
    (4): MaxPool2d(kernel_size=2, stride=2, padding=0)
    (5): ReLU()
  )
  (fc_loc): Sequential(
    (0): Linear(in_dim=90, out_dim=32)
    (1): ReLU()
    (2): Linear(in_dim=32, out_dim=6)
  )
)
```

The following is the network architecture of the first STN block inserted into the encoder of MoDem.

```
STN_Block_0_MoDem(
  (localization): Sequential(
    # By default, each image consists of three channels. Each frame in the
        ↪ observation is treated as an independent input to the STN.
    (0): Conv2d(in_channels=3, out_channels=5, kernel_size=7, stride=2)
    (1): MaxPool2d(kernel_size=4, stride=4, padding=0)
    (2): ReLU()
    (3): Conv2d(in_channels=5, out_channels=10, kernel_size=5, stride=2)
    (4): MaxPool2d(kernel_size=4, stride=4, padding=0)
    (5): ReLU()
  )
  (fc_loc): Sequential(
    (0): Linear(in_dim=90, out_dim=32)
    (1): ReLU()
    (2): Linear(in_dim=32, out_dim=6)
  )
)
```

The following is the network architecture of the second STN block inserted into the encoder of MoDem.

```
STN_Block_1_MoDem(
  (localization): Sequential(
    (0): Conv2d(in_channels=32, out_channels=8, kernel_size=7, stride=2)
    (1): MaxPool2d(kernel_size=3, stride=3, padding=0)
    (2): ReLU()
    (3): Conv2d(in_channels=8, out_channels=10, kernel_size=5, stride=2)
    (4): MaxPool2d(kernel_size=2, stride=2, padding=0)
    (5): ReLU()
  )
  (fc_loc): Sequential(
    (0): Linear(in_dim=90, out_dim=32)
    (1): ReLU()
    (2): Linear(in_dim=32, out_dim=6)
  )
)
```

# B  Environment Details

We categorize the view generalization problem into four distinct settings: novel view, moving view, shaking view, and novel FOV. In this section, we provide descriptions of the implementation details for

Table 10: **Hyperparameters for training time.**

| Hyperparameter | Value |
|---|---|
| Discount factor | 0.99 |
| Image size | $84 \times 84$ (TD-MPC) |
| | $224 \times 224$ (MoDem) |
| Frame stack | 3 (TD-MPC) |
| | 2 (MoDem) |
| Action repeat | 1 (xArm) |
| | 2 (Adroit, Finger, and Walker in DMControl) |
| | 4 (otherwise) |
| Data augmentation | $\pm 4$ pixel image shifts (TD-MPC) |
| | $\pm 10$ pixel image shifts (MoDem) |
| Seed steps | 5000 |
| Replay buffer size | Unlimited |
| Sampling technique | PER ($\alpha = 0.6$, $\beta = 0.4$) |
| Planning horizon | 5 |
| Latent dimension | 50 |
| Learning rate | 1e-3 (TD-MPC) |
| | 3e-4 (MoDem) |
| Optimizer ($\theta$) | Adam ($\beta_1 = 0.9$, $\beta_2 = 0.999$) |
| Batch size | 256 |
| Number of demos | 5 (MoDem only) |

Table 11: **Hyperparameters for test time training.**

| Hyperparameter | Value |
|---|---|
| Buffer size | 256 |
| Batch size | 32 |
| Multi-update times | 32 |
| Learning rate for encoder | 1e-6 (xArm) |
| | 1e-7 (otherwise) |
| Learning rate for STN blocks | 1e-5 |

each setting. The detailed camera settings can be referred to in the code of the environments that we are committed to releasing or in the visualization available on our website yangsizhe.github.io/MoVie.

**Novel view.** In this setting, for locomotion tasks (cheetah-run, walker-stand, walker-walk, and walker-run), the camera always faces the moving agent, while for other tasks, the camera always faces a fixed point in the environment. Therefore, as we change the camera position, the camera orientation also changes accordingly.

**Moving view.** Similar to the previous setting, the camera also always faces the moving agent or a fixed point in the environment. The camera position varies continuously.

**Shaking view.** To simulate camera shake, we applied Gaussian noise to the camera position (XYZ coordinates in meters) at each time step. For DMControl and Adroit, the mean of the distribution is 0, the standard deviation is 0.04, and we constrain the noise within the range of -0.07 to +0.07. For xArm, the mean of the distribution is 0, the standard deviation is 0.4, and we constrain the noise within the range of -0.07 to +0.07.

**Novel FOV.** We experiment with a larger FOV. For DMControl, we modify the FOV from 45 to 53. For xArm, we modify the FOV from 50 to 60. For Adroit, we modify the FOV from 45 to 50. We also experiment with a smaller FOV and results are presented in Appendix F.

## C Visualization of Feature Maps from Shallow to Deep Layers

We incorporate STN in the first two layers of the visual encoder for all tasks. After visualizing the features of different layers (as shown in Figure 7), we found that the features of shallow layers contain

more information about the spatial relationship. Therefore transforming the features of shallow layers for view generalization is reasonable.

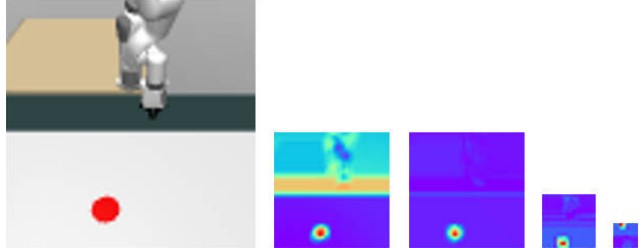

Figure 7: **Visualization of features from shallow to deep layers.** The features of shallow layers contain more information about the spatial relationship.

## D    Visualization of Feature Map Transformation

We visualize the first layer feature map of the image encoder from TD-MPC and MoVie in Figure 8. It is observed that the feature map from MoVie on the novel view exhibits a closer resemblance to that on the training view.

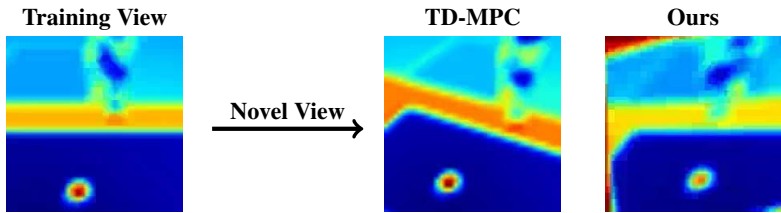

Figure 8: **Visualization of the first layer feature maps of the original encoder on the training view and on the novel view, and the learned SAE on the novel view.**

## E    Extended Description of Baselines

**TD-MPC.** We test the agent trained on training view without any adaptation in the view generalization settings.

**DM.** This is derived from MoVie by removing STN blocks, which just adapts encoder during test time.

**IDM+STN.** This is derived from MoVie by replacing the dynamics model with the inverse dynamics model which predicts the action in between based on the latent states before and after transition. The inverse dynamics model is finetuned together with the encoder and STN blocks during testing.

## F    Ablation on Different FOVs

In our main experiments, we consider the novel FOV as a FOV larger than the original. In Table 12, we present results for both smaller and larger FOV scenarios. Our method demonstrates the successful handling of both cases.

Table 12: **Ablation on different FOVs.** The best method on each setting is in **bold**.

| Cheetah-run | TD-MPC | DM | IDM+STN | MoVie |
|---|---|---|---|---|
| Small FOV | $104.85_{\pm 4.59}$ | $398.75_{\pm 17.52}$ | $75.92_{\pm 8.76}$ | $\mathbf{530.37}_{\pm \mathbf{12.84}}$ |
| Large FOV | $128.55_{\pm 6.57}$ | $379.01_{\pm 10.90}$ | $299.02_{\pm 88.47}$ | $\mathbf{532.94}_{\pm \mathbf{19.74}}$ |

# G Comparison with Other Visual RL Generalization Algorithms

In addition to baselines in the main paper, we also compare MoVie with state-of-the-art methods that focus on visual generalization or data augmentation in visual RL, *i.e.*, DrQ-v2 [35], SVEA [9], PIE-G [36], SGQN [1] and PAD [7]. Results on DMControl environments are shown in Table 13. And the training performance is shown in Table 14. It is observed that MoVie still outperforms these two methods that do not adapt in test time, while PIE-G could also achieve reasonable returns under the view perturbation. This may indicate that fusing the pre-trained networks from PIE-G into MoVie might lead to better results, which we leave as future work. Note that unlike MoVie, other methods need strong data augmentation or modification during training time.

Table 13: **Comparison with other visual RL generalization algorithms.** The best method is in **bold**.

| Task | Setting | DrQ-v2 | SVEA | SGQN | PIE-G | PAD | MoVie |
|---|---|---|---|---|---|---|---|
| Cup, catch | Novel view | $857.05_{\pm27.27}$ | $743.50_{\pm70.973}$ | $776.26_{\pm123.63}$ | $834.10_{\pm23.20}$ | $854.51_{\pm150.78}$ | $\mathbf{961.98_{\pm2.68}}$ |
| | Moving view | $911.20_{\pm1.626}$ | $843.80_{\pm28.17}$ | $\mathbf{971.08_{\pm9.30}}$ | $809.43_{\pm19.12}$ | $699.34_{\pm24.31}$ | $951.26_{\pm10.68}$ |
| | Shaking view | $638.51_{\pm39.98}$ | $501.56_{\pm66.76}$ | $943.08_{\pm7.23}$ | $869.86_{\pm14.93}$ | $870.06_{\pm21.72}$ | $\mathbf{951.20_{\pm21.01}}$ |
| | Novel FOV | $919.83_{\pm12.76}$ | $534.36_{\pm61.09}$ | $803.00_{\pm55.15}$ | $927.26_{\pm15.36}$ | $928.87_{\pm28.75}$ | $\mathbf{973.60_{\pm1.94}}$ |
| | All settings | $831.64$ | $655.80$ | $873.35$ | $860.16$ | $838.19$ | $\mathbf{959.51}$ |
| Finger, spin | Novel view | $518.75_{\pm14.63}$ | $312.38_{\pm117.76}$ | $383.78_{\pm6.82}$ | $680.10_{\pm37.47}$ | $233.60_{\pm48.93}$ | $\mathbf{892.01_{\pm1.20}}$ |
| | Moving view | $706.91_{\pm7.69}$ | $596.56_{\pm156.00}$ | $543.23_{\pm1.00}$ | $828.60_{\pm16.47}$ | $547.98_{\pm3.22}$ | $\mathbf{896.00_{\pm21.65}}$ |
| | Shaking view | $39.13_{\pm6.37}$ | $101.56_{\pm67.03}$ | $168.60_{\pm4.26}$ | $\mathbf{551.83_{\pm7.52}}$ | $209.30_{\pm7.30}$ | $284.05_{\pm473.73}$ |
| | Novel FOV | $793.70_{\pm0.65}$ | $505.03_{\pm278.99}$ | $553.26_{\pm1.38}$ | $755.86_{\pm48.25}$ | $82.08_{\pm23.4}$ | $\mathbf{917.21_{\pm1.71}}$ |
| | All settings | $514.77$ | $378.88$ | $412.21$ | $704.09$ | $215.91$ | $\mathbf{747.31}$ |

Table 14: **The performance of DrQ-v2, SVEA, SGQN, PIE-G, PAD and MoVie under the training view on 2 DMControl tasks.**

| Task | DrQ-v2 | SVEA | SGQN | PIE-G | PAD | MoVie |
|---|---|---|---|---|---|---|
| Cup, catch | $953.61_{\pm1.57}$ | $971.55_{\pm0.06}$ | $970.58_{\pm3.72}$ | $959.98_{\pm8.92}$ | $974.75_{\pm5.33}$ | $980.56_{\pm4.33}$ |
| Finger, spin | $822.4_{\pm1.02}$ | $840.33_{\pm32.08}$ | $603.10_{\pm1.17}$ | $884.96_{\pm4.30}$ | $710.26_{\pm10.92}$ | $985.20_{\pm2.25}$ |

# H Results of Original Models on Training View

The performance of the original agents without any adaptation under the training view is reported in Table 15, 16, and 17 for reference. In the context of view generalization, it is evident that the performance of agents without adaptation significantly deteriorates.

Table 15: **Training Result on DMControl.**

| Task | Cheetah, run | Walker, walk | Walker, stand | Walker, run | Cup, catch | Finger, spin |
|---|---|---|---|---|---|---|
| Reward | $658.10_{\pm9.98}$ | $944.99_{\pm21.71}$ | $983.53_{\pm5.34}$ | $697.75_{\pm11.35}$ | $980.56_{\pm4.33}$ | $985.20_{\pm2.25}$ |

| Task | Finger, turn-easy | Finger, turn-hard | Pendulum, swingup | Reacher, easy | Reacher, hard |
|---|---|---|---|---|---|
| Reward | $756.16_{\pm150.76}$ | $616.96_{\pm149.44}$ | $827.26_{\pm61.72}$ | $983.8_{\pm0.34}$ | $937.43_{\pm54.59}$ |

Table 16: **Training Result on xArm.**

| Task | Reach | Push | Peg in box | Hammer |
|---|---|---|---|---|
| Success rate (%) | $96_{\pm5}$ | $90_{\pm17}$ | $80_{\pm10}$ | $83_{\pm20}$ |

Table 17: **Training Result on Adroit.**

| Task | Door | Hammer | Pen |
|---|---|---|---|
| Success rate (%) | $96_{\pm3}$ | $78_{\pm7}$ | $48_{\pm15}$ |

