# OpenReview forum: "MoVie: Visual Model-Based Policy Adaptation for View Generalization"
_NeurIPS.cc/2023/Conference — NeurIPS 2023 poster_

### Official Review · Reviewer_BF5c · 2023-06-17

**Soundness:** 2 fair
**Presentation:** 2 fair
**Contribution:** 2 fair
**Rating:** 4
**Confidence:** 3

**Summary:**

This work presents an approach to train model-based RL methods such that it generalize to novel views on multiple RL benchmarks. The method leverages classical STN and frozen encoder to benefit the generalization performance.

**Strengths:**

The method is sound and simple with barely hyperparameters tuning

The design choices study such as different moving views and the insight are helpful for research community.

**Weaknesses:**

The writing can be largely improved on section 2 and 3. It is unclear what the model-based RL problem is, and how to define the view generalization, how to map the model to actual control/policy for evaluations, for audience that are not familiar with this line of work.

The problem is specific and the method novelty are limited.

Comparisons with baselines are unclear. At least some comparison with model-free RL and recent work [1] would be helpful.
[1] Multi-View Masked World Models for Visual Robotic Manipulation, Seo et al., 2023

**Questions:**

In the experiments, does the baseline methods see all various views in the training sets as well? I.e. is the baseline method in figure 3 also seeing different views?

There are many other choices to solve this problem. Is contrastive learning / or data augmentations on different views also a way to map different views to the same representations?

I am not sure the application on real-world robot experiments, although some visualizations are presented.

**Limitations:**

See above.

---

> ### Author Rebuttal · Authors · 2023-08-10
>
> We thank the reviewer for their constructive comments and suggestions. We address each of your comments in the following.
>
> **Q1:** The writing can be largely improved on section 2 and 3. It is unclear what the model-based RL problem is, and how to define the view generalization, how to map the model to actual control/policy for evaluations, for audience that are not familiar with this line of work.
>
> **A1:** Thank you for your suggestion. We will add more preliminaries about model-based RL in the final version. Model-based reinforcement learning is an approach to solving RL problems where the agent tries to learn a model of the environment it interacts with. Once the agent has a model of the environment, it can utilize optimal control methods, such as Model Predictive Control (MPC) and Monte Carlo Tree Search (MCTS), for planning and control. As for the definition of view generalization, we consider it as the out-of-distribution generalization about camera intrinsic and extrinsic parameters. The visual RL agent is trained on one camera parameter setting and tested on different camera parameter settings.
>
> **Q2:** The method is sound and simple with barely hyperparameters tuning. The problem is specific and the method novelty are limited.
>
> **A2:** We thank the reviewer for acknowledging the conciseness and usability of our proposed approach. We greatly value your feedback.
>
> We would like to highlight some technical contributions in our method. We made an effort to explore the prediction loss in model-based RL which can serve as a self-supervised task. We found that the dynamics model in model-based RL is a stable supervision for adaptation. We also propose spatial adaptive encode (SAE) in order to better adapt to both static and dynamic changes in view. Furthermore, our method is well-motivated, aiming to solve a fundamental and realistic question: *transforming the unseen test view to the training view*. Since we do not have paired images to align the latent space between training views and test views, we devise to use a dynamic loss combined with STN, utilizing the consistency in dynamics prediction to align the latent space.
>
> We would also like to emphasize that besides the method we propose, one main contribution in our work is the formulation of the view generalization problem and the resulting test platform, across locomotion tasks and robotic manipulation tasks.
>
> **Q3:** Comparisons with baselines are unclear. At least some comparison with model-free RL and recent work [1] would be helpful. [1] Multi-View Masked World Models for Visual Robotic Manipulation, Seo et al., 2023
>
> **A3:** Thank you for your suggestion. More comprehensive experiments which incorporate comparison with model-free RL and recent work on visual generalization can better reflect the generalization ability of our method. We compare MoVie with DrQ-v2 [1], SVEA [2] and PAD [3] on 2 tasks across 4 settings. As shown in Table 1 of the rebuttal file, MoVie outperforms other methods across all the settings. As for the recent work [4] that you mentioned, we think that its problem setup is different from ours. It uses multi-view images during training, whereas we focus on out-of-distribution generalization and use only images from single view during training.
>
> **Q4:** In the experiments, does the baseline methods see all various views in the training sets as well? I.e. is the baseline method in figure 3 also seeing different views?
>
> **A4:** No. Our method also does not see various views during training time. All the methods (ours and baselines) are trained with the single fixed view.
>
> **Q5:** There are many other choices to solve this problem. Is contrastive learning / or data augmentations on different views also a way to map different views to the same representations?
>
> **A5:** The solution to view generalization we initially thought of was contrastive learning and data augmentation. In order to learn a view invariant representation, images from different views or strong data augmentation are needed. However, using images from different views (as shown in Figure 3 of the rebuttal file) or strong data augmentation negatively impacts the performance of the RL algorithm [5]. And this is the reason why we abandoned contrastive learning and data augmentation.
>
> **Q6:** I am not sure the application on real-world robot experiments, although some visualizations are presented.
>
> **A6:**  We believe that our manuscript clearly demonstrates that online adaptation to view changes in practical robotic scenarios, covering an extensive range of robotic manipulation tasks to validate its applicability. Our setting is under a practical setting, suitable for both simulation and real world (single view training and any view adaptation). In our future work, we would like to conduct experiments in real world.
>
> [1] Denis Yarats, Rob Fergus, Alessandro Lazaric, and Lerrel Pinto. Mastering visual continuous control: Improved data-augmented reinforcement learning. arXiv preprint arXiv:2107.09645, 2021.
>
> [2] Nicklas Hansen, Hao Su, and Xiaolong Wang. Stabilizing deep q-learning with convnets and vision 362 transformers under data augmentation. Advances in Neural Information Processing Systems, 34, 2021.
>
> [3] Nicklas Hansen, Rishabh Jangir, Yu Sun, Guillem Alenyà, Pieter Abbeel, Alexei A Efros, Lerrel Pinto, and Xiaolong Wang. Self-supervised policy adaptation during deployment. ICLR, 2021.
>
> [4] Multi-View Masked World Models for Visual Robotic Manipulation, Seo et al., 2023
>
> [5] Kostrikov, Ilya, Denis Yarats, and Rob Fergus. "Image augmentation is all you need: Regularizing deep reinforcement learning from pixels." arXiv preprint arXiv:2004.13649 (2020).

---

> ### Author Response · Authors · 2023-08-19
> **Thank you for the review and awaiting your response**
>
> We sincerely thank you for your efforts in reviewing our paper and the suggestions again.
>
> We believe that we have resolved all the concerns mentioned in the review. Should there be any additional concerns, we are more than happy to address them! Thank you very much!

---

### Official Review · Reviewer_E44Y · 2023-07-06

**Soundness:** 3 good
**Presentation:** 3 good
**Contribution:** 2 fair
**Rating:** 6
**Confidence:** 3

**Summary:**

This paper mainly provides a training paradigm. Utilizing the dynamic transition model of the environment during the testing phase as a supervisory signal, STN is used to quickly finetune the mapping of observed potential states, resulting in better performance of the strategy mapping trained on a single view task on unseen test views. This work conducted comparative experiments on three types of tasks and four challenging views, achieving results that surpass existing methods. Ablation experiments were conducted to prove the effectiveness of each module setting.

**Strengths:**

The training paradigm provided in this article can effectively handle the generalization problem when view changes occur, and a thorough comparative experiment and ablation experiment have been conducted on the proposed model. The core idea of reconstructing mapping h using the environmental dynamic transfer model in the paper is effective.

**Weaknesses:**

1. The subscripts for o and a in Formula 2 are missing.
2. There are issues with the baseline selection of IDM+STN. Cheetah run alone cannot prove that IDM+STN is generally better than IDM. It is not a problem to conduct ablation experiments on the Cheetah run model alone, but this cannot be used as sufficient evidence for the superiority of IDM+STN over IDM and thus selecting it as the baseline.
3. Chapter 3 emphasizes the differences between Formula 2 and Formula 1, on one hand, due to the fixed parameters of Network d, and on the other hand due to the use of SAE in Network h. One of the core methods is to fix the parameters of d, which is intuitive to avoid changes in the target domain of h during the update process, resulting in the failure of the policy pi. However, the ablation experiment showed that the setting of d^* did not achieve a consistently effective effect.
4. The main approach is to use the dynamic transfer model of the environment under the new view as supervision to quickly train the changed h '. Can you obtain a generalized h directly based on o'?

**Questions:**

1. Have the training methods of the three baselines in the comparative experiment also been finetuned? How much data was used during finetuning, and is this amount of data acceptable for practical applications?
2. Can the network of h ^ SAE be directly used in training to achieve learning from multiple perspectives of tasks? Directly generalize from an unseen perspective without finetuning.
3. The reason why Movie is better than IDM+STN should be explained.

**Limitations:**

The authors have discussed the limitations

---

> ### Author Rebuttal · Authors · 2023-08-10
>
> We thank the reviewer for their constructive comments and suggestions. We address each of your comments in the following.
>
> **Q1:** The subscripts for o and a in Formula 2 are missing.
>
> **A1:** Thank you for catching this. We will fix this in the final version.
>
> **Q2:** There are issues with the baseline selection of IDM+STN. Cheetah run alone cannot prove that IDM+STN is generally better than IDM. It is not a problem to conduct ablation experiments on the Cheetah run model alone, but this cannot be used as sufficient evidence for the superiority of IDM+STN over IDM and thus selecting it as the baseline.
>
> **A2:** We appreciate your feedback and acknowledge your concerns regarding our choice of IDM+STN as the baseline. We broaden our experimental setup to include 5 tasks across 4 settings. The results provided in Table 3 of the rebuttal file shows that IDM+STN is generally better than IDM.
>
> **Q3:** Chapter 3 emphasizes the differences between Formula 2 and Formula 1, on one hand, due to the fixed parameters of Network d, and on the other hand due to the use of SAE in Network h. One of the core methods is to fix the parameters of d, which is intuitive to avoid changes in the target domain of h during the update process, resulting in the failure of the policy pi. However, the ablation experiment showed that the setting of d^* did not achieve a consistently effective effect.
>
> **A3:** We agree with the reviewer that the pursuit of a universally optimal representation is critical. But it is inherently challenging for a single approach to excel across all tasks in the field of reinforcement learning. In our experiments, although finetuning the dynamics model slightly outperforms our method in the novel FOV scenario, it underperforms in others. Furthermore, we conducted experiments on more tasks and settings, and the results presented in Table 4 of the rebuttal file demonstrate fixing the dynamics model during adaptation is generally better than finetuning it.
>
> **Q4:** The main approach is to use the dynamic transfer model of the environment under the new view as supervision to quickly train the changed h '. Can you obtain a generalized h directly based on o'?
>
> **A4:** In order to learn a view invariant representation and directly generalize to an unseen view without finetuning, images from different views are needed. However, as shown in Figure 3 of the rebuttal file, training with multi-view images leads to poor training performance in our attempt. Additionally, accessing images from different views is not easy, especially in the real world.
>
> **Q5:** Have the training methods of the three baselines in the comparative experiment also been finetuned?  How much data was used during finetuning, and is this amount of data acceptable for practical applications?
>
> **A5:** TD-MPC in the baselines has not been finetuned at test time. DM and IDM+STN have been finetuned for fair comparison. We ran 20 episodes on each of the settings and we found that the performance has been greatly improved after adaptation of one episode in most tasks. An episode consists of tens or hundreds of steps in our robotic tasks, which is quite acceptable for practical applications.
>
> **Q6:** Can the network of h ^ SAE be directly used in training to achieve learning from multiple perspectives of tasks? Directly generalize from an unseen perspective without finetuning.
>
> **A6:** Please see A4 for details.
>
> **Q7:** The reason why Movie is better than IDM+STN should be explained.
>
> **A7:** Intuitively, IDM and DM are somewhat similar, while there are two reasons that make IDM loss worse than DM under our setting:
>
> 1. Model-based RL methods such as TD-MPC and MoDem do not introduce IDM during training originally, which is mainly because introducing IDM loss is not generally helping as shown in Figure 4 of the rebuttal file. This is because additional optimization objectives could heavily affect optimization outcomes.
>
> 2. Directly using IDM during test time (as our baseline) results in inconsistency between training and test, which thus leads to suboptimal performance. As shown in Table 1, Table2, Table 3, Table 4 and Table 5 in our main paper, despite being slightly better than MoVie on a few tasks, IDM+STN underperforms MoVie on most tasks.

---

> > ### Comment · Reviewer_E44Y · 2023-08-21
> >
> > I appreciate the authors' response. They addressed most of my concerns. I decide to raise my rating score.

---

> ### Author Response · Authors · 2023-08-19
> **Thank you for the review and awaiting your response**
>
> We sincerely thank you for your efforts in reviewing our paper and the suggestions again.
>
> We believe that we have resolved all the concerns mentioned in the review. Should there be any additional concerns, we are more than happy to address them! Thank you very much!

---

### Official Review · Reviewer_N1Ux · 2023-07-06

**Soundness:** 3 good
**Presentation:** 4 excellent
**Contribution:** 2 fair
**Rating:** 6
**Confidence:** 4

**Summary:**

The paper addresses the novel problem of view generalization in reinforcement learning, where an RL agent is trained on an environment with a fixed view and then evaluated on a test environment having the exact same dynamics but observed from a different perspective. In order to address this issue, the authors introduce an innovative method that permits test-time adaptation to the new view.
The authors integrate a learnable spatial transformer network (STN) into the feature extraction component of the agent. This augmented encoder is then fine-tuned at test time to generate features that enable a frozen latent dynamic model to predict future state representations. An essential aspect of the authors' approach is that this adaptation does not necessitate any reward signal, which is a key advantage.
Experimental results demonstrate a significant reduction in the generalization gap presented by a new view across several heterogeneous benchmarks.




**Strengths:**

The paper is well-structured and easy to follow, providing clear and detailed explanations that make  believe that I could readily reproduce the method based on the descriptions provided.
The authors introduce a new problem in the field that could stimulate further research.
The proposed method's strength lies in its test-time adaptation capability, which eliminates the need for any reward signal, making it highly applicable to real-world scenarios. This characteristic also suggests that the approach could be integrated into any model-based method, enhancing its universality.
Another advantage is the apparent minimal interaction required with the testing environment, promoting efficiency.
The method is evaluated across various environments, with the results showcasing impressive improvements compared to non-adaptive methods.
The paper's ablation study effectively illustrates the contribution of each component of the method, highlighting the importance of every aspect in achieving the observed results.  Additionally, the annex offers excellent visualizations demonstrating the impact of the spatial transformer on the feature map, providing intuitive understanding of the method's mechanics.



**Weaknesses:**

Despite the many strengths of the paper, a few areas could benefit from further development and clarification.

1. It would be beneficial to see a comparison of the performance decrease relative to the original view. The lack of this data makes it challenging to truly gauge the significance of the generalization gap and the efficacy of the proposed method.

2. The scope of comparative studies is somewhat limited. The paper primarily compares with methods not designed for visual adaptation, which may not provide the most insightful comparison. It would have been advantageous to see how the proposed approach stacks up against other strategies specifically intended for visual adaptation.

3. An exploration of whether the proposed method can also benefit model-free algorithms such as Soft Actor-Critic (SAC) is missing. This could broaden the applicability of the findings and provide additional insights into the method's versatility.

4. It remains unclear why the proposed method underperforms with the Inverse Dynamics Model (IDM). A deeper exploration of this anomaly could bolster the robustness and reliability of the approach.

5. Finally, the similarity between the proposed method and the PAD method, which performs test-time domain adaptation in model-free RL, raises questions. The key differences are the use of a Spatial Transformer Network and the leveraging of the latent dynamics model of model-based methods instead of adding it as an auxiliary component. However, these differences, while noteworthy, do not necessarily represent a substantial departure from the PAD method. This similarity begs the question of whether the presented method offers significant novelty or if it is essentially an adjustment of existing approaches.

**Questions:**

1. To better understand the importance of the generalization gap, could you provide performance data related to the original view? This information would help quantify the performance decrease when the agent is switched to a different view.

2. The use of inverse dynamics models with spatial transformer networks in your method isn't entirely clear.
Does IDM completely replace the DM in the TD-MPC algorithm?
Or does it just used as a side network during TD-MPC training and used for test adaptation only?


**Limitations:**

The authors mentioned a limitation of this work: the proposed method has not been tested in real-world scenarios, such as with physical robots, which may limit its immediate applicability.
Another potential limitation could be that the method has only been tested on model-based reinforcement learning (RL), whereas it would be relatively straightforward to evaluate its effectiveness in model-free RL.

---

> ### Author Rebuttal · Authors · 2023-08-10
>
> We thank the reviewer for their constructive comments and suggestions. We address each of your comments in the following.
>
> **Q1:** It would be beneficial to see a comparison of the performance decrease relative to the original view. The lack of this data makes it challenging to truly gauge the significance of the generalization gap and the efficacy of the proposed method.
>
> **A1:** The performance of the original agents without any adaptation under the training view has been reported in Table 13, 14, and 15 in appendix of the original paper. When tested on our view generalization testing platform, the performance of the agent without adaptation experiences a significant decrease.
>
> **Q2:** It would have been advantageous to see how the proposed approach stacks up against other strategies specifically intended for visual adaptation.
>
> **A2:** We appreciate your suggestion on including other visual adaptation methods in baselines. And we compare MoVie with PAD [1] which also adapts visual encoder at test time on 2 tasks across 4 settings (We choose tasks on which PAD performs well at training time, and the addition of auxiliary tasks negatively affects PAD's performance on some other tasks.). The results are provided in Table 1 of the rebuttal file. It is observed that MoVie outperforms PAD significantly across all 8 settings.
>
> **Q3:** An exploration of whether the proposed method can also benefit model-free algorithms such as Soft Actor-Critic (SAC) is missing. This could broaden the applicability of the findings and provide additional insights into the method's versatility.
>
> **A3:** We believe this is an exciting direction and we made initial attempt toward this direction.  We attempted to apply our adaptation method to DrQ-v2 [2] by integrating a dynamics model and SAE (spatial adaptive encoder) during testing while it is not able to achieve reasonable results. In our initial attempt, the view generalization ability has not been greatly improved, and the performance gets even worse after adaptation on some tasks as shown in Table 5 of the rebuttal file. We believe it requires non-trivial effort to explore how to use more self-supervised losses for model-free RL methods, but we want to point out that this is not trivial to find the suitable loss that does not hurt the training but also helps during test time.
>
> **Q4:** It remains unclear why the proposed method underperforms with the Inverse Dynamics Model (IDM). A deeper exploration of this anomaly could bolster the robustness and reliability of the approach.
>
> **A4:** Intuitively, IDM and DM are somewhat similar, while there are two reasons that make IDM loss worse than DM under our setting:
>
> 1. Model-based RL methods such as TD-MPC and MoDem do not introduce IDM during training originally, which is mainly because introducing IDM loss is not generally helping as shown in Figure 4 of the rebuttal file. This is because additional optimization objectives could heavily affect optimization outcomes.
>
> 2. Directly using IDM during test time (as our baseline) results in inconsistency between training and test, which thus leads to suboptimal performance. As shown in Table 1, Table2, Table 3, Table 4 and Table 5 in our main paper, despite being slightly better than MoVie on a few tasks, IDM+STN underperforms MoVie on most tasks.
>
> **Q5:** The similarity between the proposed method and the PAD method, which performs test-time domain adaptation in model-free RL, raises questions. This similarity begs the question of whether the presented method offers significant novelty or if it is essentially an adjustment of existing approaches.
>
> **A5:** Although one component of our method is similar to PAD [1], we have made significant improvements:
>
> 1. As shown in Figure 4 of the rebuttal file, auxiliary tasks added at training time like PAD could negatively impact the performance of the algorithm. And our method does not require any modification at training time.
>
> 2. And we made an effort to explore the prediction loss in model-based RL which can serve as a self-supervised task. We found that the dynamics model in model-based RL is a stable supervision for adaptation.
>
> 3. Our method is well-motivated, aiming to solve a fundamental and realistic challenge: *generalize by transforming the unseen test view to the training view*. Since we do not have paired images to align the latent space between training views and test views, we devise to use a dynamic loss combined with STN, utilizing the consistency in dynamics prediction to align the latent space.
>
> 4. Furthermore, as you said, we propose spatial adaptive encode (SAE) in order to better adapt to both static and dynamic changes in view.
>
> **Q6:** To better understand the importance of the generalization gap, could you provide performance data related to the original view?
>
> **A6:** Please see A1 for details.
>
> **Q7:** The use of inverse dynamics models with spatial transformer networks in your method isn't entirely clear. Does IDM completely replace the DM in the TD-MPC algorithm? Or does it just used as a side network during TD-MPC training and used for test adaptation only?
>
> **A7:** The IDM is added and trained at test time and the loss of inverse dynamics is used to optimize IDM and SAE (spatial adaptive encoder) simultaneously. The reason we didn't add IDM as auxiliary tasks during training is that it would negatively impact the training performance as shown in Figure 4 of the rebuttal file. Thanks for your thoughtful question and we will add more details about IDM in the final version.
>
> [1] Nicklas Hansen, Rishabh Jangir, Yu Sun, Guillem Alenyà, Pieter Abbeel, Alexei A Efros, Lerrel Pinto, and Xiaolong Wang. Self-supervised policy adaptation during deployment. ICLR, 2021.
>
> [2] Denis Yarats, Rob Fergus, Alessandro Lazaric, and Lerrel Pinto. Mastering visual continuous control: Improved data-augmented reinforcement learning. arXiv preprint arXiv:2107.09645, 2021.

---

> > ### Comment · Reviewer_N1Ux · 2023-08-11
> >
> > I appreciate the authors' responses and clarifications regarding my questions.
> > After considering their input, I still believe that this paper is of sufficient quality to be accepted for NeurIPS.

---

> > > ### Author Response · Authors · 2023-08-11
> > >
> > > We thank the reviewer for acknowledging our efforts.

---

### Official Review · Reviewer_CdSm · 2023-07-06

**Soundness:** 2 fair
**Presentation:** 3 good
**Contribution:** 2 fair
**Rating:** 5
**Confidence:** 4

**Summary:**

This paper focuses on improving the generalization ability of visual DRL to adapt to unseen views. The authors propose a model-based policy adaptation approach that combines spatial transformer networks with a self-supervised dynamics prediction objective to address this problem. The effectiveness of the approach is evaluated through experiments on three commonly used benchmarks.

**Strengths:**

- This paper is well-written and easy to follow.
- The proposed method is well-motivated.
- The authors try to solve a realistic and essential problem, i.e., view generalization, especially in real-world or sim-to-real robotic scenarios.

**Weaknesses:**

- While the problem setting is attractive, the technical contribution is insufficient as a full NeurIPS paper. It is a straightforward combination of multiple existing works, like STN.
- There needs to be more than the current evaluation to demonstrate the superiority over existing works.

**Questions:**

- The experiments need to be more comprehensive. The comparison baselines used are more like ablation studies rather than state-of-the-art (SOTA) methods, such as TD-MPC (the backbone of MoVie), DM (MoVie without STN), and IDM+STN (MoVie replacing DM with IDM). The authors should consider incorporating more commonly-used or SOTA data augmentation algorithms for visual DRL, such as DrQ-v2[1] and SEVA[2].
- The authors mention multiple times that only shallow STNs can improve performance, but they fail to provide deeper analysis, such as visual illustrations. Is the number of layers set for all tasks, or do different tasks require their own suitable layers? This information would be valuable.
- The authors claim that "Our proposed method enables direct deployment of offline or simulation-trained agents ...", However, the lack of practical experimental demonstrations, such as robot manipulation, makes their claim unconvincing. I strongly suggest that the authors conduct real-world robot experiments to assess the applicability of their proposed method.

References:
- [1] Denis Yarats, Rob Fergus, Alessandro Lazaric, and Lerrel Pinto. Mastering visual continuous control: Improved data-augmented reinforcement learning. arXiv preprint arXiv:2107.09645, 2021.
- [2] Nicklas Hansen, Hao Su, and Xiaolong Wang. Stabilizing deep q-learning with convnets and vision 362 transformers under data augmentation. Advances in Neural Information Processing Systems, 34, 2021.

**Limitations:**

Not Applicable.

---

> ### Author Rebuttal · Authors · 2023-08-10
>
> We thank the reviewer for their constructive comments and suggestions. We address each of your comments in the following.
>
> **Q1:** While the problem setting is attractive, the technical contribution is insufficient as a full NeurIPS paper. It is a straightforward combination of multiple existing works, like STN.
>
> **A1:** We would like to respectfully point out that our method is not just combining simple techniques straightforwardly. Instead, it is well-motivated, aiming to solve a fundamental and realistic problem: *generalize by transforming the unseen test view to the training view*. Though such motivation is simple, it is not trivial to do this since we do not store the training views, and thus we do not have paired images to align the latent space between training views and test views. To solve this, we devise to use a dynamic loss combined with STN, utilizing the consistency in dynamics prediction to align the latent space.
>
> Moreover, one main contribution of our paper is to formulate and study the view generalization problem in RL systematically, across locomotion and manipulation tasks, covering a broad range of tasks. Our proposed method also achieves non-trivial success across all these tasks over strong baselines.
>
> We hope the reviewer acknowledges the efforts we made in addressing the realistic view generalization problem.
>
>
> **Q2:** There needs to be more than the current evaluation to demonstrate the superiority over existing works.
>
> **A2:** Please see A3 for details.
>
> **Q3:** The experiments need to be more comprehensive. The comparison baselines used are more like ablation studies rather than state-of-the-art (SOTA) methods, such as TD-MPC (the backbone of MoVie), DM (MoVie without STN), and IDM+STN (MoVie replacing DM with IDM). The authors should consider incorporating more commonly-used or SOTA data augmentation algorithms for visual DRL, such as DrQ-v2 [1] and SVEA [2].
>
> **A3:** Thank you for your suggestion. To further show the advantage of MoVie, we compare MoVie with DrQ-v2 [1], SVEA [2] and PAD [3] on 2 tasks across 4 settings. As shown in Table 1 of the rebuttal file, MoVie outperforms other methods across all these settings. By the way, the training performance of PAD limited the scale of the experiments. We trained these baselines on several tasks, but the performance of PAD can compete with others on only 2 tasks.
>
> **Q4:** The authors mention multiple times that only shallow STNs can improve performance, but they fail to provide deeper analysis, such as visual illustrations. Is the number of layers set for all tasks, or do different tasks require their own suitable layers? This information would be valuable.
>
> **A4:** We incorporate STN in the first two layers of the visual encoder for all tasks. After visualizing the features of different layers (as shown in Figure 1 of the rebuttal file), we found that the features of shallow layers contain more information about the spatial relationship; therefore transforming the features of shallow layers for view generalization is reasonable.
>
> **Q5:** The authors claim that "Our proposed method enables direct deployment of offline or simulation-trained agents ...", However, the lack of practical experimental demonstrations, such as robot manipulation, makes their claim unconvincing. I strongly suggest that the authors conduct real-world robot experiments to assess the applicability of their proposed method.
>
> **A5:** We agree that real-world evaluation is valuable. However, simulation remains critical to the community for a number of reasons: (i) it provides researchers with common benchmarks that accurately measure progress in the area, (ii) it facilitates reproducibility and statistically significant results, and (iii) improves equity in the area of  machine learning by removing barriers of entry for researchers to contribute to our collective knowledge. While we agree that reliable real-world benchmarks and evaluations would be valuable and very welcome, it should not be treated as a prerequisite for research on visual generalization due to it still being an open and underexplored problem.
>
>
>
> [1] Denis Yarats, Rob Fergus, Alessandro Lazaric, and Lerrel Pinto. Mastering visual continuous control: Improved data-augmented reinforcement learning. arXiv preprint arXiv:2107.09645, 2021.
>
> [2] Nicklas Hansen, Hao Su, and Xiaolong Wang. Stabilizing deep q-learning with convnets and vision 362 transformers under data augmentation. Advances in Neural Information Processing Systems, 34, 2021.
>
> [3] Nicklas Hansen, Rishabh Jangir, Yu Sun, Guillem Alenyà, Pieter Abbeel, Alexei A Efros, Lerrel Pinto, and Xiaolong Wang. Self-supervised policy adaptation during deployment. ICLR, 2021.

---

> > ### Comment · Reviewer_CdSm · 2023-08-19
> >
> > I appreciate the authors' additional experimental results and critical analysis. They addressed most of my concerns. So I decide to raise my final rating score. However, Regarding Q5, I acknowledge the importance of simulation but have concerns about your claim that "Our proposed method enables direct deployment of offline or simulation-trained agents" (Lines 186-189). Due to the significant difference (domain gap) in input images between the simulation and the real world, a sim-2-real module is usually needed to mitigate these discrepancies. I am skeptical about the feasibility of deploying your MoVie directly to real-world experiments. Thus, I recommend that the authors present concrete experimental findings instead of mere assertions.

---

> > > ### Author Response · Authors · 2023-08-20
> > >
> > > Thank you for your constructive response and the acknowledgement of our effort.
> > >
> > > We agree that there is a difference in input images between the simulation and the real world; hence, real world experiment is beneficial. We would like to point out that our method can narrow the domain gap in input images to some extent. As shown in the table below, MoVie effectively facilitates the agent's adaptation to different visual domains. In this experiment, we introduce changes to the appearance of objects such as the ground, table, and background in the environment and report cumulative rewards for DMControl tasks and success rates for robotic manipulation tasks. While these being a surrogate setting, we also plan to conduct experiments in real world in our future work to further demonstrate that MoVie enables direct deployment of offline or simulation-trained agents.
> > >
> > > | Tasks | TD-MPC (original)  | TD-MPC under appearance change | MoVie under appearance change |
> > > | :-----: | :-----: | :-----: | :-----: |
> > > | Walker, walk |  $ 944.99\pm21.71 $  |  $ 589.76\pm53.27 $  |  $ 882.00\pm72.68 $  |
> > > | xArm, reach |  $ 0.96\pm0.05 $  |  $ 0.15\pm0.05 $  |  $ 0.78\pm0.02 $  |

---

> ### Author Response · Authors · 2023-08-19
> **Thank you for the review and awaiting your response**
>
> We sincerely thank you for your efforts in reviewing our paper and the suggestions again.
>
> We believe that we have resolved all the concerns mentioned in the review. Should there be any additional concerns, we are more than happy to address them! Thank you very much!

---

### Official Review · Reviewer_CADg · 2023-07-09

**Soundness:** 3 good
**Presentation:** 4 excellent
**Contribution:** 3 good
**Rating:** 6
**Confidence:** 4

**Summary:**

Adapting policy into new view setup is an important task in RL. This paper presents MoVie, Model-based policies for View generalization to achieve the fast view adaptation of model-based policy. Specifically, they combine spatial transformer networks into the encoder models and train it during test-time using dynamics prediction loss. Despite the simplicity, the experimental results show that the strong performance of MoVie over several benchmarks on various test set.

**Strengths:**

1. The proposed method is simple yet provide stable performance gain over various methods. Experimental validation is extensive enough.
2. The paper is well written and easy to follow.

**Weaknesses:**

1. Technical novelty and technical depth is limited. I might decrease scores if I found works that combine STN with test-time training.

Besides, few discussions are made on the choice of test-time loss and entire architecture. For example, the experimental results suggest that the DM is better than IDM especially when combining test-time training, but the paper lacks discussion on why it is. Besides, I agree prediction loss is one of the natural loss to train STN, but I think it is also possible to extend the method to model-free methods by using some self-supervised loss function (like contrastive learning).

2. I think the experimental validation is already comprehensive, but it lacks qualitative analysis of their representation before/after adaptation.

3. No theoretical justification for the choice of test-time loss and or entire architecture.


**Questions:**

See weakness section.

---

> ### Author Rebuttal · Authors · 2023-08-10
>
> We thank the reviewer for their constructive comments and suggestions. We address each of your comments in the following.
>
> **Q1:** The proposed method is simple yet provide stable performance gain over various methods. Experimental validation is extensive enough. But Technical novelty and technical depth is limited. I might decrease scores if I found works that combine STN with test-time training.
>
> **A1:** We thank the reviewer for acknowledging the effectiveness of our proposed approach. We emphasize that besides the method we propose, one main contribution in our work is the **formulation** of the view generalization problem and the resulting test platform, across locomotion tasks and robotic manipulation tasks.
>
> **Q2:**  Few discussions are made on the choice of test-time loss and entire architecture. For example, the experimental results suggest that the DM is better than IDM especially when combining test-time training, but the paper lacks discussion on why it is.
>
> **A2:**  The main reason we use the dynamic loss during test time is because such a loss is common in model-based RL. Hence, using the same objective during test time leads to consistency between training and test. One might naturally think that IDM could be similar to our DM objective, while there are two reasons that make IDM loss worse than DM under our setting:
>
> 1. Model-based RL methods such as TD-MPC and MoDem do not introduce IDM during training originally, which is mainly because introducing IDM loss is not generally helping as shown in Figure 4 of the rebuttal file. This is because additional optimization objectives could heavily affect optimization outcomes.
>
> 2. Directly using IDM during test time (as our baseline) results in inconsistency between training and test, which thus leads to suboptimal performance. As shown in Table 1, Table2, Table 3, Table 4 and Table 5 in our main paper, despite being slightly better than MoVie on a few tasks, IDM+STN underperforms MoVie on most tasks.
>
> **Q3:** I agree prediction loss is one of the natural loss to train STN, but I think it is also possible to extend the method to model-free methods by using some self-supervised loss function (like contrastive learning)
>
> **A3:** We believe this is an exciting direction and we made initial attempt toward this direction. Some previous works attempted to include auxiliary tasks during training, such as PAD [1], but it was observed that these auxiliary tasks could negatively impact the training performance of the RL algorithm. We attempted to apply our adaptation method in DrQ-v2 [2] while it is not able to achieve reasonable results as shown in Table 5 of the rebuttal file. In our initial attempt, the view generalization ability has not been greatly improved, and the performance gets even worse after adaptation on some tasks. We believe it requires non-trivial effort to explore how to use more self-supervised losses for model-free RL methods, but we want to point out that this is not trivial to find the suitable loss that does not hurt the training but also helps during test time.
>
> **Q4:** I think the experimental validation is already comprehensive, but it lacks qualitative analysis of their representation before/after adaptation
>
> **A4:** In Appendix C (also in Figure 2 of the rebuttal file), we visualize the features of shallow layers before and after adaptation, and it is observed that features are transformed after adaptation, closer to the training view. This interprets why our method could adapt to different views.
>
> **Q5:** No theoretical justification for the choice of test-time loss and or entire architecture
>
> **A5:**
> The main focus of our work is to study and solve the view generalization problem in RL, more from an empirical perspective. We also agree about the importance of rigorous theoretical sketch, which could be our future exploration direction.
>
> [1] Nicklas Hansen, Rishabh Jangir, Yu Sun, Guillem Alenyà, Pieter Abbeel, Alexei A Efros, Lerrel Pinto, and Xiaolong Wang. Self-supervised policy adaptation during deployment. ICLR, 2021.
>
> [2] Denis Yarats, Rob Fergus, Alessandro Lazaric, and Lerrel Pinto. Mastering visual continuous control: Improved data-augmented reinforcement learning. arXiv preprint arXiv:2107.09645, 2021.

---

> > ### Comment · Reviewer_CADg · 2023-08-20
> > **I would like to keep my rating**
> >
> > Thank you for your detailed response. I still think the paper provides a good contribution to the community, and therefore keep my rating.

---

> ### Author Response · Authors · 2023-08-19
> **Thank you for the review and awaiting your response**
>
> We sincerely thank you for your efforts in reviewing our paper and the suggestions again.
>
> We believe that we have resolved all the concerns mentioned in the review. Should there be any additional concerns, we are more than happy to address them! Thank you very much!

---

### Author Rebuttal · Authors · 2023-08-10

We thank all the reviewers for their insightful comments and suggestions.

We are delighted to receive your recognition of the strengths in our work, including but not limited to the meaningful problem formulation, well-motivated and effective method, extensive experimental validation and good writing – “stable performance gain over various methods” “experimental validation is extensive enough” (CADg), “solve a realistic and essential problem” “the paper is well written and easy to follow” (CdSm), “introduce a new problem in the field that could stimulate further research” (N1Ux), “effectively handle the generalization problem when view changes occur” (E44Y), “the method is sound and simple” (BF5c).

Your suggestions and concerns are also valuable. We have replied separately and conducted extensive additional experiments as support. The experiment results are given in the PDF file.

**EXP1: Comparison with other generalization or data augmentation algorithms for visual RL** in reply to Reviewer CdSm, Reviewer N1Ux and Reviewer BF5c. Results are given in Table 1. MoVie outperforms other methods including DrQ-v2 [1], SVEA [2] and PAD [3] across all the settings.

**EXP2: Ablation for baseline selection** in reply to Reviewer E44Y. Results are given in Table 3 and Table 4. We broaden our experimental setup to include 5 tasks across 4 settings. The results show that IDM+STN is generally better than IDM and fixing the dynamics model during adaptation is generally better than finetuning it.

**EXP3: Visualizations of features from shallow to deep layers** in reply to Reviewer CdSm. Results are given in Figure 1. We found that the features of shallow layers contain more information about the spatial relationship, so transforming the features of shallow layers for view generalization is reasonable.

**EXP4: Visualizations of features before and after adaptation** in reply to Reviewer CADg. Results are given in Figure 2. It is observed that features are transformed after adaptation closer to the training view. This interprets why our method could adapt to different views.

**EXP5: Comparison of DrQ-v2 and DrQ-v2 with our adaptation method** in reply to Reviewer CADg and Reviewer N1Ux. Results are given in Table 5. While extending our method to model-free RL is exciting, the view generalization ability has not been greatly improved in our initial attempt. We want to point out that it is not trivial to find the suitable loss that does not hurt the training but also helps during test time.

**EXP6: Training performance of DrQ-v2 using single-view and multi-view images during training** in reply to Reviewer N1Ux and Reviewer E44Y. Results are given in Figure 3. It is observed that training with multi-view images leads to poor training performance.

**EXP7: Training performance of TD-MPC with and without IDM as auxiliary task** in reply to Reviewer N1Ux and Reviewer E44Y. Results are given in Figure 4. It is observed that auxiliary tasks added at training time could negatively impact the performance.

[1] Denis Yarats, Rob Fergus, Alessandro Lazaric, and Lerrel Pinto. Mastering visual continuous control: Improved data-augmented reinforcement learning. arXiv preprint arXiv:2107.09645, 2021.

[2] Nicklas Hansen, Hao Su, and Xiaolong Wang. Stabilizing deep q-learning with convnets and vision 362 transformers under data augmentation. Advances in Neural Information Processing Systems, 34, 2021.

[3] Nicklas Hansen, Rishabh Jangir, Yu Sun, Guillem Alenyà, Pieter Abbeel, Alexei A Efros, Lerrel Pinto, and Xiaolong Wang. Self-supervised policy adaptation during deployment. ICLR, 2021.

---

### Decision · Program_Chairs · 2023-09-21

**Decision:**

Accept (poster)

**Comment:**

This submission deals with the view generalization problem in image-based reinforcement leanring. While the overall rating of this submission was fair, it had a few critical feedbacks such as its limited technical contribution and experimental justimation. Due to these reasons, a few reviewers and the AC had a discussion. The conlcusion is that positives, like the simplicity of the method without much hyperparameter tuning and also its interesting problem setup, are worth enough to be shared with other audience in the community. We however strongly suggest the authors to update the submission with more evaluations and results to solidify the paper.